# Stabilizing Deep $Q$-Learning with ConvNets and Vision Transformers under Data Augmentation

**Nicklas Hansen**[1]  **Hao Su**[1]  **Xiaolong Wang**[1]
[1]University of California, San Diego
nihansen@ucsd.edu {haosu,xiw012}@eng.ucsd.edu

## Abstract

While agents trained by Reinforcement Learning (RL) can solve increasingly challenging tasks directly from visual observations, generalizing learned skills to novel environments remains very challenging. Extensive use of data augmentation is a promising technique for improving generalization in RL, but it is often found to decrease sample efficiency and can even lead to divergence. In this paper, we investigate causes of instability when using data augmentation in common off-policy RL algorithms. We identify two problems, both rooted in high-variance $Q$-targets. Based on our findings, we propose a simple yet effective technique for stabilizing this class of algorithms under augmentation. We perform extensive empirical evaluation of image-based RL using both ConvNets and Vision Transformers (ViT) on a family of benchmarks based on DeepMind Control Suite, as well as in robotic manipulation tasks. Our method greatly improves stability and sample efficiency of ConvNets under augmentation, and achieves generalization results competitive with state-of-the-art methods for image-based RL in environments with unseen visuals. We further show that our method scales to RL with ViT-based architectures, and that data augmentation may be especially important in this setting.[†]

## 1 Introduction

Reinforcement Learning (RL) from visual observations has achieved tremendous success in various applications such as video-games [43, 4, 70], robotic manipulation [37], and autonomous navigation [42, 83]. However, it is still very challenging for current methods to generalize the learned skills to novel environments, and policies trained by RL can easily overfit to the training environment [81, 13], especially for high-dimensional observation spaces such as images [8, 58].

Increasing the variability in training data via domain randomization [66, 50] and data augmentation [57, 35, 33, 51] has demonstrated encouraging results for learning policies invariant to changes in environment observations. Specifically, recent works on data augmentation [35, 33] both show improvements in sample efficiency from simple cropping and translation augmentations, but the studies also conclude that additional data augmentation in fact *decrease* sample efficiency and even cause divergence. While these augmentations have the potential to improve generalization, the increasingly varied data makes the optimization more challenging and risks instability. Unlike supervised learning, balancing the trade-off between stability and generalization in RL requires substantial trial and error.

In this paper, we illuminate causes of instability when applying data augmentation to common off-policy RL algorithms [43, 38, 15, 18]. Based on our findings, we provide an intuitive method for stabilizing this class of algorithms under use of strong data augmentation. Specifically, we find two main causes of instability in previous work's application of data augmentation: (i) indiscriminate application of data augmentation resulting in high-variance $Q$-targets; and (ii) that $Q$-value estimation strictly from augmented data results in over-regularization.

---

[†]Website and code is available at: https://nicklashansen.github.io/SVEA.

To address these problems, we propose **SVEA**: **S**tabilized $Q$-**V**alue **E**stimation under **A**ugmentation, a simple yet effective framework for data augmentation in off-policy RL that greatly improves stability of $Q$-value estimation. Our method consists of the following three components: Firstly, by only applying augmentation in $Q$-value estimation of the *current* state, *without* augmenting $Q$-targets used for bootstrapping, SVEA circumvents erroneous bootstrapping caused by data augmentation; Secondly, we formulate a modified $Q$-objective that optimizes $Q$-value estimation jointly over both augmented and unaugmented copies of the observations; Lastly, for SVEA implemented with an actor-critic algorithm, we optimize the actor strictly on unaugmented data, and instead learn a generalizable policy indirectly through parameter-sharing. Our framework can be implemented efficiently without additional forward passes nor introducing additional learnable parameters.

We perform extensive empirical evaluation on the DeepMind Control Suite [64] and extensions of it, including the DMControl Generalization Benchmark [21] and the Distracting Control Suite [60], as well as a set of robotic manipulation tasks. Our method greatly improve $Q$-value estimation with ConvNets under a set of strong data augmentations, and achieves sample efficiency, asymptotic performance, and generalization that is competitive or better than previous state-of-the-art methods in all tasks considered, at a lower computational cost. Finally, we show that our method scales to RL with Vision Transformers (ViT) [10]. We find that ViT-based architectures are especially prone to overfitting, and data augmentation may therefore be a key component for large-scale RL.

## 2 Related Work

**Representation Learning.** Learning visual invariances using data augmentation and self-supervised objectives has proven highly successful in computer vision [46, 45, 82, 74, 68, 65, 75, 27, 7]. For example, Chen et al. [7] perform an extensive study on data augmentation (e.g. random cropping and image distortions) for contrastive learning, and show that representations pre-trained with such transformations transfer effectively to downstream tasks. While our work also uses data augmentation for learning visual invariances, we leverage the $Q$-objective of deep $Q$-learning algorithms instead of auxiliary representation learning tasks.

**Visual Learning for RL.** Numerous methods have been proposed with the goal of improving sample efficiency [29, 56, 68, 76, 40, 59, 61, 54, 77] of image-based RL. Recently, using self-supervision to improve generalization in RL has also gained interest [80, 47, 55, 1, 22, 21, 72]. Notably, Zhang et al. [80] and Agarwal et al. [1] propose to learn behavioral similarity embeddings via auxiliary tasks (bisimulation metrics and contrastive learning, respectively), and Hansen et al. [21] learn visual invariances through an auxiliary prediction task. While these results are encouraging, it has also been shown in [29, 40, 22, 79, 41] that the best choice of auxiliary tasks depends on the particular RL task, and that joint optimization with sub-optimally chosen tasks can lead to gradient interference. We achieve competitive sample-efficiency and generalization results *without* the need for carefully chosen auxiliary tasks, and our method is therefore applicable to a larger variety of RL tasks.

**Data Augmentation and Randomization for RL.** Our work is directly inspired by previous work on generalization in RL by domain randomization [66, 50, 48, 52, 6] and data augmentation [36, 9, 71, 35, 33, 51, 61, 21]. For example, Tobin et al. [66] show that a neural network trained for object localization in a simulation with randomized visual augmentations improves real world generalization. Similarly, Lee et al.[36] show that application of a random convolutional layer to observations during training improve generalization in 3D navigation tasks. More recently, extensive studies on data augmentation [35, 33] have been conducted with RL, and conclude that, while small random crops and translations can improve sample efficiency, most data augmentations *decrease* sample efficiency and cause divergence. We illuminate main causes of instability, and propose a framework for data augmentation in deep $Q$-learning algorithms that drastically improves stability and generalization.

**Improving Deep $Q$-Learning.** While deep $Q$-learning algorithms such as Deep $Q$-Networks (DQN) [43] have achieved impressive results in image-based RL, the temporal difference objective is known to have inherent instabilities when used in conjunction with function approximation and off-policy data [63]. Therefore, a variety of algorithmic improvements have been proposed to improve convergence [24, 73, 25, 23, 53, 38, 15, 14, 28]. For example, Hasselt et al. [24] reduce overestimation of $Q$-values by decomposing the target $Q$-value estimation into action selection and action evaluation using separate networks. Lillicrap et al. [38] reduce target variance by defining the target $Q$-network as a slow-moving average of the online $Q$-network. Our method also improves $Q$-value estimation, but we specifically address the instability of deep $Q$-learning algorithms on augmented data.

# 3 Preliminaries

**Problem formulation.** We formulate the interaction between environment and policy as a Markov Decision Process (MDP) [2] $\mathcal{M} = \langle \mathcal{S}, \mathcal{A}, \mathcal{P}, r, \gamma \rangle$, where $\mathcal{S}$ is the state space, $\mathcal{A}$ is the action space, $\mathcal{P}: \mathcal{S} \times \mathcal{A} \mapsto \mathcal{S}$ is the state transition function that defines a conditional probability distribution $\mathcal{P}(\cdot|\mathbf{s}_t, \mathbf{a}_t)$ over all possible next states given a state $\mathbf{s}_t \in \mathcal{S}$ and action $\mathbf{a}_t \in \mathcal{A}$ taken at time $t$, $r: \mathcal{S} \times \mathcal{A} \mapsto \mathbb{R}$ is a reward function, and $\gamma \in [0, 1)$ is the discount factor. Because image observations only offer partial state observability [30], we define a state $\mathbf{s}_t$ as a sequence of $k + 1$ consecutive frames $(\mathbf{o}_t, \mathbf{o}_{t-1}, \ldots, \mathbf{o}_{t-k})$, $\mathbf{o} \in \mathcal{O}$, where $\mathcal{O}$ is the high-dimensional image space, as proposed in Mnih et al. [43]. The goal is then to learn a policy $\pi: \mathcal{S} \mapsto \mathcal{A}$ that maximizes discounted return $R_t = \mathbb{E}_{\Gamma \sim \pi}[\sum_{t=1}^{T} \gamma^t r(\mathbf{s}_t, \mathbf{a}_t)]$ along a trajectory $\Gamma = (\mathbf{s}_0, \mathbf{s}_1, \ldots, \mathbf{s}_T)$ obtained by following policy $\pi$ from an initial state $\mathbf{s}_0 \in \mathcal{S}$ to a state $\mathbf{s}_T$ with state transitions sampled from $\mathcal{P}$, and $\pi$ is parameterized by a collection of learnable parameters $\theta$. For clarity, we hereon generically denote parameterization with subscript, e.g. $\pi_\theta$. We further aim to learn parameters $\theta$ s.t. $\pi_\theta$ generalizes well (i.e., obtains high discounted return) to unseen MDPs, which is generally unfeasible without further assumptions about the structure of the space of MDPs. In this work, we focus on generalization to MDPs $\overline{\mathcal{M}} = \langle \overline{\mathcal{S}}, \mathcal{A}, \mathcal{P}, r, \gamma \rangle$, where states $\overline{\mathbf{s}}_t \in \overline{\mathcal{S}}$ are constructed from observations $\overline{\mathbf{o}}_t \in \overline{\mathcal{O}}$, $\mathcal{O} \subseteq \overline{\mathcal{O}}$ of a *perturbed* observation space $\overline{\mathcal{O}}$ (e.g. unseen visuals), and $\overline{\mathcal{M}} \sim \mathbb{M}$ for a space of MDPs $\mathbb{M}$.

**Deep $Q$-Learning.** Common model-free off-policy RL algorithms aim to estimate an optimal state-action value function $Q^*: \mathcal{S} \times \mathcal{A} \mapsto \mathbb{R}$ as $Q_\theta(\mathbf{s}, \mathbf{a}) \approx Q^*(\mathbf{s}, \mathbf{a}) = \max_{\pi_\theta} \mathbb{E}[R_t | \mathbf{s}_t = \mathbf{s}, \mathbf{a}_t = \mathbf{a}]$ using function approximation. In practice, this is achieved by means of the single-step Bellman residual $\left(r(\mathbf{s}_t, \mathbf{a}_t) + \gamma \max_{\mathbf{a}'_t} Q_\psi^{\text{tgt}}(\mathbf{s}_{t+1}, \mathbf{a}'_t)\right) - Q_\theta(\mathbf{s}_t, \mathbf{a}_t)$ [62], where $\psi$ parameterizes a *target* state-action value function $Q^{\text{tgt}}$. We can choose to minimize this residual (also known as the *temporal difference* error) directly wrt $\theta$ using a mean squared error loss, which gives us the objective

$$\mathcal{L}_Q(\theta, \psi) = \mathbb{E}_{\mathbf{s}_t, \mathbf{a}_t, \mathbf{s}_{t+1} \sim \mathcal{B}} \left[ \frac{1}{2} \left[ \left( r(\mathbf{s}_t, \mathbf{a}_t) + \gamma \max_{\mathbf{a}'_t} Q_\psi^{\text{tgt}}(\mathbf{s}_{t+1}, \mathbf{a}'_t) \right) - Q_\theta(\mathbf{s}_t, \mathbf{a}_t) \right]^2 \right], \quad (1)$$

where $\mathcal{B}$ is a replay buffer with transitions collected by a behavioral policy [39]. From here, we can derive a greedy policy directly by selecting actions $\mathbf{a}_t = \arg\max_{\mathbf{a}_t} Q_\theta(\mathbf{s}_t, \mathbf{a}_t)$. While $Q^{\text{tgt}} = Q$ and periodically setting $\psi \longleftarrow \theta$ exactly recovers the objective of DQN [43], several improvements have been proposed to improve stability of Eq. 1, such as Double Q-learning [24], Dueling $Q$-networks [73], updating target parameters using a slow-moving average of the online $Q$-network [38]:

$$\psi_{n+1} \longleftarrow (1 - \zeta)\psi_n + \zeta\theta_n \quad (2)$$

for an iteration step $n$ and a momentum coefficient $\zeta \in (0, 1]$, and others [25, 23, 53, 14, 28]. As computing $\max_{\mathbf{a}'_t} Q_\psi^{\text{tgt}}(\mathbf{s}_{t+1}, \mathbf{a}'_t)$ in Eq. 1 is intractable for large and continuous action spaces, a number of prominent *actor-critic* algorithms that additionally learn a policy $\pi_\theta(\mathbf{s}_t) \approx \arg\max_{\mathbf{a}_t} Q_\theta(\mathbf{s}_t, \mathbf{a}_t)$ have therefore been proposed [38, 15, 18].

**Soft Actor-Critic** (SAC) [18] is an off-policy actor-critic algorithm that learns a state-action value function $Q_\theta$ and a stochastic policy $\pi_\theta$ (and optionally a temperature parameter), where $Q_\theta$ is optimized using a variant of the objective in Eq. 1 and $\pi_\theta$ is optimized using a $\gamma$-discounted maximum-entropy objective [84]. To improve stability, SAC is also commonly implemented using Double Q-learning and the slow-moving target parameters from Eq. 2. We will in the remainder of this work describe our method in the context of a generic off-policy RL algorithm that learns a parameterized state-action value function $Q_\theta$, while we in our experiments discussed in Section 6 evaluate of our method using SAC as base algorithm.

# 4 Pitfalls of Data Augmentation in Deep $Q$-Learning

In this section, we aim to illuminate the main causes of instability from naïve application of data augmentation in $Q$-value estimation. Our goal is to learn a $Q$-function $Q_\theta$ for an MDP $\mathcal{M}$ that generalizes to novel MDPs $\overline{\mathcal{M}} \sim \mathbb{M}$ with unseen visuals, and we leverage data augmentation as an optimality-invariant state transformation $\tau$ to induce a bisimulation relation [34, 17] between a state $\mathbf{s}$ and its transformed (augmented) counterpart $\mathbf{s}^{\text{aug}} = \tau(\mathbf{s}, \nu)$ with parameters $\nu \sim \mathcal{V}$.

**Definition 1** (Optimality-Invariant State Transformation [33])**.** *Given an MDP $\mathcal{M}$, a state transformation $\tau: \mathcal{S} \times \mathcal{V} \mapsto \mathcal{S}$ is an optimality-invariant state transformation if $Q(\mathbf{s}, \mathbf{a}) = Q(\tau(\mathbf{s}, \nu), \mathbf{a}) \ \forall \mathbf{s} \in \mathcal{S}, \mathbf{a} \in \mathcal{A}, \nu \in \mathcal{V}$, where $\nu \in \mathcal{V}$ parameterizes the transformation $\tau$.*

Following our definitions of $\mathcal{M}, \overline{\mathcal{M}}$ from Section 3, we can further extend the concept of optimality-invariant transformations to MDPs, noting that a change of state space (e.g. perturbed visuals) itself can be described as a transformation $\overline{\tau}\colon \mathcal{S} \times \overline{\mathcal{V}} \mapsto \overline{\mathcal{S}}$ with unknown parameters $\overline{\nu} \in \overline{\mathcal{V}}$. If we choose the set of parameters $\mathcal{V}$ of a state transformation $\tau$ to be sufficiently large such that it intersects with $\overline{\mathcal{V}}$ with high probability, we can therefore expect to improve generalization to state and observation spaces not seen during training. However, while naïve application of data augmentation as in previous work [35, 33, 61, 54] may potentially improve generalization, it can be harmful to $Q$-value estimation. We hypothesize that this is primarily because it dramatically increases the size of the observed state space, and consequently also increases variance $\mathrm{Var}\,[Q(\tau(\mathbf{s}, \nu))] \geq \mathrm{Var}\,[Q(\mathbf{s})]$, $\nu \sim \mathcal{V}$ when $\mathcal{V}$ is large. Concretely, we identify the following two issues:

**Pitfall 1: Non-deterministic $Q$-target.** For deep $Q$-learning algorithms, previous work [35, 33, 61, 54] applies augmentation to both state $\mathbf{s}_t^{\mathrm{aug}} \triangleq \tau(\mathbf{s}_t, \nu)$ and successor state $\mathbf{s}_{t+1}^{\mathrm{aug}} \triangleq \tau(\mathbf{s}_{t+1}, \nu')$ where $\nu, \nu' \sim \mathcal{V}$. Compared with DQN [43] that uses a deterministic (more precisely, periodically updated) $Q$-target, this practice introduces a non-deterministic $Q$-target $r(\mathbf{s}_t, \mathbf{a}_t) + \gamma \max_{\mathbf{a}_t'} Q_\psi^{\mathrm{tgt}}(\mathbf{s}_{t+1}^{\mathrm{aug}}, \mathbf{a}_t')$ depending on the augmentation parameters $\nu'$. As observed in the original DQN paper, high-variance target values are detrimental to $Q$-learning algorithms, and may cause divergence due to the "deadly triad" of function approximation, bootstrapping, and off-policy learning [63]. This motivates the work to introduce a slowly changing target network, and several other works have refined the $Q$-target update rule [38, 15] to further reduce volatility. However, because data augmentation is inherently non-deterministic, it can greatly increase variance in $Q$-target estimation and exacerbates the issue of volatility, as shown in Figure 1 (top). This is particularly troubling in actor-critic algorithms such as DDPG [38] and SAC [18], where the $Q$-target is estimated from $(\mathbf{s}_{t+1}, \mathbf{a}')$, $\mathbf{a}' \sim \pi(\cdot|\mathbf{s}_{t+1})$, which introduces an additional source of error from $\pi$ that is non-negligible especially when $\mathbf{s}_{t+1}$ is augmented.

**Pitfall 2: Over-regularization.** Data augmentation was originally introduced in the supervised learning regime as a regularizer to prevent overfitting of high-capacity models. However, for RL, even learning a policy in the training environment is hard. While data augmentation may improve generalization, it greatly increases the difficulty of policy learning, i.e., optimizing $\theta$ for $Q_\theta$ and potentially a behavior network $\pi_\theta$. Particularly, when the temporal difference loss from Eq. 1 cannot be well minimized, the large amount of augmented states dominate the gradient, which significantly impacts $Q$-value estimation of both augmented and unaugmented states. We refer to this issue as *over-regularization* by data augmentation. Figure 1 (bottom) shows the mean difference in $Q$-predictions made with augmented vs. unaugmented data in fully converged DrQ [33] agents trained with *shift* augmentation. Augmentations such as affine-jitter, random convolution, and random overlay incur large differences in estimated $Q$-values. While such difference can be reduced by regularizing the optimization with each individual augmentation, we emphasize that even the minimal shift augmentation used throughout training incurs non-zero difference. Since $\psi$ is commonly chosen to be a

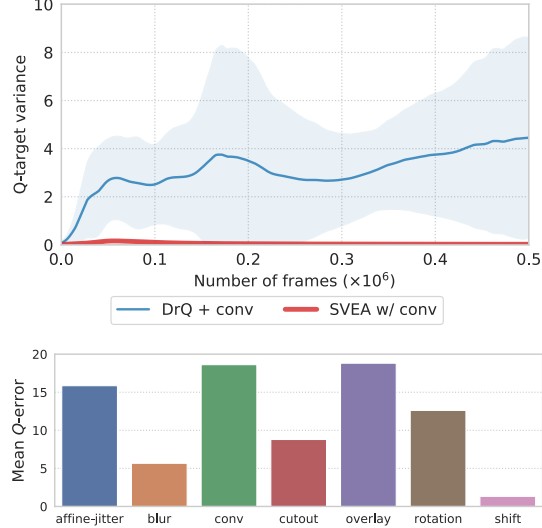

*Figure 1. (Top)* **Mean $Q$-target variance** of DrQ [33] and SVEA (ours), both trained with *conv* augmentation [36]. *(Bottom)* **Mean difference in $Q$-value estimation on augmented vs. non-augmented data.** We measure mean absolute error in $Q$-value estimation from converged DrQ agents (trained with *shift* augmentation) on the same observations before and after augmentation. Both figures are averages across 5 seeds for each of the 5 tasks from DMControl-GB.

moving average of $\theta$ as in Eq. 2, such differences caused by over-regularization affect $Q_\theta$ and $Q_\psi^{\mathrm{tgt}}$ equally, and optimization may therefore still diverge depending on the choice of data augmentation. As such, there is an inherent trade-off between accurate $Q$-value estimation and generalization when using data augmentation. In the following section, we address these pitfalls.

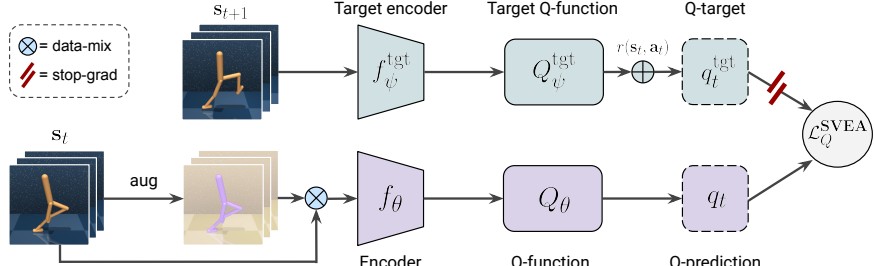

*Figure 2.* **Overview.** An observation $\mathbf{s}_t$ is transformed by data augmentation $\tau(\cdot, \nu)$, $\nu \sim \mathcal{V}$ to produce a view $\mathbf{s}_t^{\text{aug}}$. The $Q$-function $Q_\theta$ is then jointly optimized on both augmented and unaugmented data wrt the objective in Eq. 7, with the $Q$-target of the Bellman equation computed from an unaugmented observation $\mathbf{s}_{t+1}$. We illustrate our data-mixing strategy by the $\otimes$ operator.

## 5 Method

We propose **SVEA**: **S**tabilized $Q$-**V**alue **E**stimation under **A**ugmentation, a general framework for visual generalization in RL by use of data augmentation. SVEA applies data augmentation in a novel learning framework leveraging two data streams – with and without augmented data, respectively. Our method is compatible with any standard off-policy RL algorithm without changes to the underlying neural network that parameterizes the policy, and it requires no additional forward passes, auxiliary tasks, nor learnable parameters. While SVEA in principle does not make any assumptions about the structure of states $\mathbf{s}_t \in \mathcal{S}$, we here describe our method in the context of image-based RL.

### 5.1 Architectural Overview

An overview of the SVEA architecture is provided in Figure 2. Our method leverages properties of common neural network architectures used in off-policy RL without introducing additional learnable parameters. We subdivide the neural network layers and corresponding learnable parameters of a state-action value function into sub-networks $f_\theta$ (denoted the state *encoder*) and $Q_\theta$ (denoted the *Q-function*) s.t $q_t \triangleq Q_\theta(f_\theta(\mathbf{s}_t), \mathbf{a}_t)$ is the predicted $Q$-value corresponding to a given state-action pair $(\mathbf{s}_t, \mathbf{a}_t)$. We similarly define the target state-action value function s.t. $q_t^{\text{tgt}} \triangleq r(\mathbf{s}_t, \mathbf{a}_t) + \gamma \max_{\mathbf{a}'} Q_\psi^{\text{tgt}}(f_\psi^{\text{tgt}}(\mathbf{s}_{t+1}), \mathbf{a}')$ is the target $Q$-value for $(\mathbf{s}_t, \mathbf{a}_t)$, and we define parameters $\psi$ as an exponential moving average of $\theta$ as in Eq. 2. Depending on the choice of underlying algorithm, we may choose to additionally learn a parameterized policy $\pi_\theta$ that shares encoder parameters with $Q_\theta$ and selects actions $\mathbf{a}_t \sim \pi_\theta(\cdot|f_\theta(\mathbf{s}_t))$.

To circumvent erroneous bootstrapping from augmented data (as discussed in Section 4), we strictly apply data augmentation in $Q$-value estimation of the *current* state $\mathbf{s}_t$, *without* applying data augmentation to the successor state $\mathbf{s}_{t+1}$ used in Eq. 1 for bootstrapping with $Q_\psi^{\text{tgt}}$ (and $\pi_\theta$ if applicable), which addresses Pitfall 1. If $\pi_\theta$ is learned (i.e., SVEA is implemented with an actor-critic algorithm), we also optimize it strictly from unaugmented data. To mitigate over-regularization in optimization of $f_\theta$ and $Q_\theta$ (Pitfall 2), we further employ a modified $Q$-objective that leverages both augmented and unaugmented data, which we introduce in the following section.

### 5.2 Learning Objective

Our method redefines the temporal difference objective from Eq. 1 to better leverage data augmentation. First, recall that $q_t^{\text{tgt}} = r(\mathbf{s}_t, \mathbf{a}_t) + \gamma \max_{\mathbf{a}'} Q_\psi^{\text{tgt}}(f_\psi^{\text{tgt}}(\mathbf{s}_{t+1}), \mathbf{a}')$. Instead of learning to predict $q_t^{\text{tgt}}$ only from state $\mathbf{s}_t$, we propose to minimize a nonnegative linear combination of $\mathcal{L}_Q$ over two individual data streams, $\mathbf{s}_t$ and $\mathbf{s}_t^{\text{aug}} = \tau(\mathbf{s}_t, \nu)$, $\nu \sim \mathcal{V}$, which we define as the objective

$$\mathcal{L}_Q^{\text{SVEA}}(\theta, \psi) \triangleq \alpha \mathcal{L}_Q\left(\mathbf{s}_t, q_t^{\text{tgt}}; \theta, \psi\right) + \beta \mathcal{L}_Q\left(\mathbf{s}_t^{\text{aug}}, q_t^{\text{tgt}}; \theta, \psi\right) \tag{3}$$

$$= \mathbb{E}_{\mathbf{s}_t, \mathbf{a}_t, \mathbf{s}_{t+1} \sim \mathcal{B}}\left[\alpha \left\|Q_\theta(f_\theta(\mathbf{s}_t), \mathbf{a}_t) - q_t^{\text{tgt}}\right\|_2^2 + \beta \left\|Q_\theta(f_\theta(\mathbf{s}_t^{\text{aug}}), \mathbf{a}_t) - q_t^{\text{tgt}}\right\|_2^2\right], \tag{4}$$

where $\alpha, \beta$ are constant coefficients that balance the ratio of the unaugmented and augmented data streams, respectively, and $q_t^{\text{tgt}}$ is computed strictly from unaugmented data. $\mathcal{L}_Q^{\text{SVEA}}(\theta, \psi)$ serves as a *data-mixing* strategy that oversamples unaugmented data as an implicit variance reduction technique.

As we will verify empirically in Section 6, data-mixing is a simple and effective technique for variance reduction that works well in tandem with our proposed modifications to bootstrapping. For $\alpha = \beta$, the objective in Eq. 4 can be evaluated in a single, batched forward-pass by rewriting it as:

$$\mathbf{g}_t = [\mathbf{s}_t, \tau(\mathbf{s}_t, \nu)]_N \tag{5}$$

$$h_t = \left[q_t^{\text{tgt}}, q_t^{\text{tgt}}\right]_N \tag{6}$$

$$\mathcal{L}_Q^{\mathbf{SVEA}}(\theta, \psi) = \mathbb{E}_{\mathbf{s}_t, \mathbf{a}_t, \mathbf{s}_{t+1} \sim \mathcal{B},\, \nu \sim \mathcal{V}} \left[ (\alpha + \beta) \| Q_\theta(f_\theta(\mathbf{g}_t), \mathbf{a}_t) - h_t \|_2^2 \right], \tag{7}$$

where $[\cdot]_N$ is a concatenation operator along the batch dimension $N$ for $\mathbf{s}_t, \mathbf{s}_t^{\text{aug}} \in \mathbb{R}^{N \times C \times H \times W}$ and $q_t^{\text{tgt}} \in \mathbb{R}^{N \times 1}$, which is illustrated as $\otimes$ in Figure 2. Empirically, we find $\alpha = 0.5, \beta = 0.5$ to be both effective and practical to implement, which we adopt in the majority of our experiments. However, more sophisticated schemes for selecting $\alpha, \beta$ and/or varying them as training progresses could be interesting directions for future research. If the base algorithm of choice learns a policy $\pi_\theta$, its objective $\mathcal{L}_\pi(\theta)$ is optimized solely on unaugmented states $\mathbf{s}_t$ without changes to the objective, and a stop-grad operation is applied after $f_\theta$ to prevent non-stationary gradients of $\mathcal{L}_\pi(\theta)$ from interfering with $Q$-value estimation, i.e., only the objective from Eq. 4 or optionally Eq. 7 updates $f_\theta$ using stochastic gradient descent. As described in Section 5.1, parameters $\psi$ are updated using an exponential moving average of $\theta$ and a stop-grad operation is therefore similarly applied after $Q_\psi^{\text{tgt}}$. We summarize our method for $\alpha = \beta$ applied to a generic off-policy algorithm in Algorithm 1.

---

**Algorithm 1** Generic **SVEA** off-policy algorithm (▶ naïve augmentation, ▶ our modifications)

---

$\theta, \theta_\pi, \psi$: randomly initialized network parameters, $\psi \longleftarrow \theta$      $\triangleright$ Initialize $\psi$ to be equal to $\theta$
$\eta, \zeta$: learning rate and momentum coefficient
$\alpha, \beta$: loss coefficients, *default:* $(\alpha = 0.5, \beta = 0.5)$
1: **for** timestep $t = 1...T$ **do**
    **act:**
2:      $\mathbf{a}_t \sim \pi_\theta\left(\cdot | f_\theta(\mathbf{s}_t)\right)$      $\triangleright$ Sample action from policy
3:      $\mathbf{s}_t' \sim \mathcal{P}(\cdot | \mathbf{s}_t, \mathbf{a}_t)$      $\triangleright$ Sample transition from environment
4:      $\mathcal{B} \leftarrow \mathcal{B} \cup (\mathbf{s}_t, \mathbf{a}_t, r(\mathbf{s}_t, \mathbf{a}_t), \mathbf{s}_t')$      $\triangleright$ Add transition to replay buffer
    **update:**
5:      $\{\mathbf{s}_i, \mathbf{a}_i, r(\mathbf{s}_i, \mathbf{a}_i), \mathbf{s}_i' \mid i = 1...N\} \sim \mathcal{B}$      $\triangleright$ Sample batch of transitions
6:      $\mathbf{s}_i = \tau(\mathbf{s}_i, \nu_i),\ \mathbf{s}_i' = \tau(\mathbf{s}_i', \nu_i'),\ \nu_i, \nu_i' \sim \mathcal{V}$      ▶ Naïve application of data augmentation
7:      **for** transition $i = 1..N$ **do**
8:          $\theta_\pi \longleftarrow \theta_\pi - \eta \nabla_{\theta_\pi} \mathcal{L}_\pi(\mathbf{s}_i; \theta_\pi)$ (if applicable)      $\triangleright$ Optimize $\pi_\theta$ with SGD
9:          $q_i^{\text{tgt}} = r(\mathbf{s}_i, \mathbf{a}_i) + \gamma \max_{\mathbf{a}_i'} Q_\psi^{\text{tgt}}(f_\psi^{\text{tgt}}(\mathbf{s}_i'), \mathbf{a}_i')$      $\triangleright$ Compute $Q$-target
10:          $\mathbf{s}_i^{\text{aug}} = \tau(\mathbf{s}_i, \nu_i),\ \nu_i \sim \mathcal{V}$      ▶ Apply stochastic data augmentation
11:          $\mathbf{g}_i = \left[\mathbf{s}_i, \mathbf{s}_i^{\text{aug}}\right]_N,\ h_i = \left[q_i^{\text{tgt}}, q_i^{\text{tgt}}\right]_N$      ▶ Pack data streams
12:          $\theta \longleftarrow \theta - \eta \nabla_\theta \mathcal{L}_Q^{\mathbf{SVEA}}(\mathbf{g}_i, h_i; \theta, \psi)$      ▶ Optimize $f_\theta$ and $Q_\theta$ with SGD
13:          $\psi \longleftarrow (1 - \zeta)\psi + \zeta\theta$      $\triangleright$ Update $\psi$ using EMA of $\theta$

---

## 6 Experiments

We evaluate both sample efficiency, asymptotic performance, and generalization of our method and a set of strong baselines using both Con­vNets and Vision Transformers (ViT) [10] in tasks from DeepMind Control Suite (DMCon­trol) [64] as well as a set of robotic manip­ulation tasks. DMControl offers challenging and diverse continuous control tasks and is widely used as a benchmark for image-based

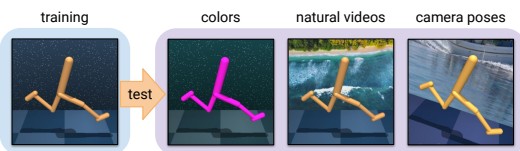

*Figure 3.* **Experimental setup.** Agents are trained in a fixed environment and are expected to general­ize to novel environments with e.g. random colors, backgrounds, and camera poses.

RL [19, 20, 76, 59, 35, 33]. To evaluate generalization of our method and baselines, we test methods under challenging distribution shifts (as illustrated in Figure 3) from the DMControl Gen­eralization Benchmark (DMControl-GB) [21], the Distracting Control Suite (DistractingCS) [60], as well as distribution shifts unique to the robotic manipulation environment. Code is available at https://github.com/nicklashansen/dmcontrol-generalization-benchmark.

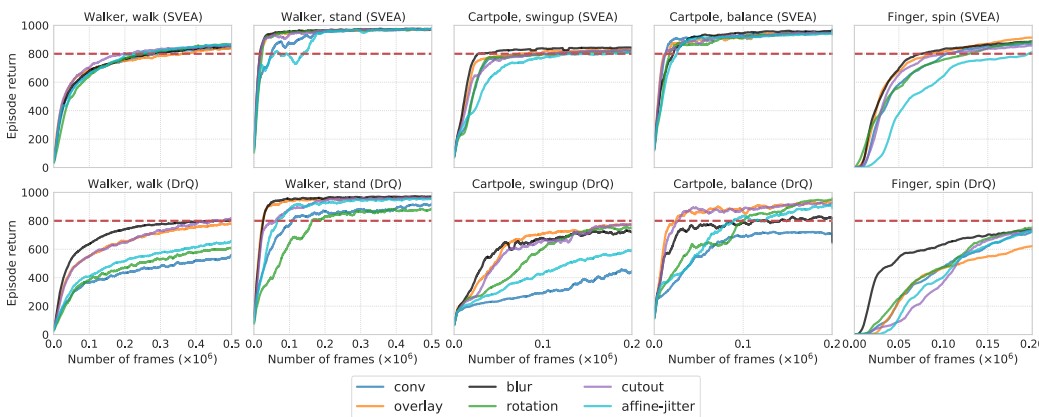

*Figure 4.* **Data augmentations.** Training performance of SVEA (top) and DrQ (bottom) under 6 common data augmentations. Mean of 5 seeds. Red line at 800 return is for visual guidance only. We omit visualization of std. deviations for clarity, but provide per-augmentation comparisons to DrQ (including std. deviations) across all tasks in Appendix B, and test performances in Appendix C.

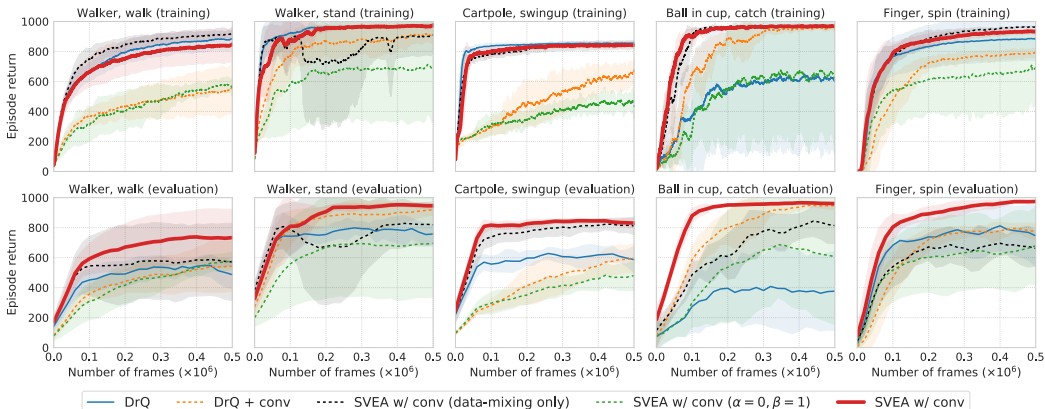

*Figure 5.* **Training and test performance.** We compare SVEA to DrQ with and without random convolution augmentation, as well as a set of ablations. *Data-mixing only* indiscriminately applies our data-mixing strategy to all data streams, and $(\alpha = 0, \beta = 1)$ only augments $Q$-predictions but without data-mixing. We find both components to contribute to SVEA's success. *Top:* episode return on the training environment during training. *Bottom:* generalization measured by episode return on the `color_hard` benchmark of DMControl-GB. Mean of 5 seeds, shaded area is $\pm 1$ std. deviation.

**Setup.** We implement our method and baselines using SAC [18] as base algorithm, and we apply random shift augmentation to all methods by default. This makes our base algorithm equivalent to DrQ [33] when K=1,M=1; we refer to the base algorithm as *unaugmented* and consider stability under additional data augmentation. We use the **same** network architecture and hyperparameters for **all** methods (whenever applicable), and adopt the setup from Hansen and Wang [21]. Observations are stacks of 3 RGB frames of size $84 \times 84 \times 3$ (and $96 \times 96 \times 3$ in ViT experiments). In the DMControl-GB and DistractingCS benchmarks, all methods are trained for 500k frames and evaluated on all 5 tasks from DMControl-GB used in prior work, and we adopt the same experimental setup for robotic manipulation. See Appendix H for hyperparameters and further details on our experimental setup.

**Baselines and data augmentations.** We benchmark our method against the following strong baselines: (1) **CURL** [59], a contrastive learning method for RL; (2) **RAD** that applies a random crop; (3) **DrQ** that applies a random shift; (4) **PAD** [22] that adapts to test environments using self-supervision; (5) **SODA** [21] that applies data augmentation in auxiliary learning; as well as a number of ablations. We compare to the K=1,M=1 setting of DrQ by default, but also provide comparison to varying $K, M$. We experiment with a diverse set of data augmentations proposed in previous work on RL and computer vision, namely random *shift* [33], random convolution (denoted *conv*) [36], random *overlay* [21], random *cutout* [9], Gaussian *blur*, random *affine-jitter*, and random *rotation* [35, 16]. We provide samples for all data augmentations in Appendix C and test environments in Appendix E.

*Table 1.* **Comparison to state-of-the-art.** Test performance (episode return) of methods trained in a single, fixed environment and evaluated on (i) randomized colors, and (ii) natural video backgrounds from DMControl-GB. Results for CURL, RAD, PAD, and SODA are obtained from [21] and we report mean and std. deviation over 5 seeds. DrQ corresponds to our SAC base algorithm using random shift augmentation. SVEA matches or outperforms prior methods in all tasks considered.

| DMControl-GB (random colors) | CURL | RAD | DrQ | PAD | SODA (conv) | SODA (overlay) | **SVEA** (conv) | **SVEA** (overlay) |
|---|---|---|---|---|---|---|---|---|
| walker, walk | 445 ±99 | 400 ±61 | 520 ±91 | 468 ±47 | 697 ±66 | 692 ±68 | **760** ±**145** | 749 ±61 |
| walker, stand | 662 ±54 | 644 ±88 | 770 ±71 | 797 ±46 | 930 ±12 | 893 ±12 | **942** ±**26** | 933 ±24 |
| cartpole, swingup | 454 ±110 | 590 ±53 | 586 ±52 | 630 ±63 | 831 ±21 | 805 ±28 | **837** ±**23** | 832 ±23 |
| ball_in_cup, catch | 231 ±92 | 541 ±29 | 365 ±210 | 563 ±50 | 892 ±37 | 949 ±19 | **961** ±**7** | 959 ±5 |
| finger, spin | 691 ±12 | 667 ±154 | 776 ±134 | 803 ±72 | 901 ±51 | 793 ±128 | **977** ±**5** | 972 ±6 |

| DMControl-GB (natural videos) | CURL | RAD | DrQ | PAD | SODA (conv) | SODA (overlay) | **SVEA** (conv) | **SVEA** (overlay) |
|---|---|---|---|---|---|---|---|---|
| walker, walk | 556 ±133 | 606 ±63 | 682 ±89 | 717 ±79 | 635 ±48 | 768 ±38 | 612 ±144 | **819** ±**71** |
| walker, stand | 852 ±75 | 745 ±146 | 873 ±83 | 935 ±20 | 903 ±56 | 955 ±13 | 795 ±70 | **961** ±**8** |
| cartpole, swingup | 404 ±67 | 373 ±72 | 485 ±105 | 521 ±76 | 474 ±76 | 758 ±62 | 606 ±85 | **782** ±**27** |
| ball_in_cup, catch | 316 ±119 | 481 ±26 | 318 ±157 | 436 ±55 | 539 ±111 | **875** ±**56** | 659 ±110 | 871 ±106 |
| finger, spin | 502 ±19 | 400 ±64 | 533 ±119 | 691 ±80 | 363 ±185 | 695 ±97 | 764 ±86 | **808** ±**33** |

*Figure 6. (Left)* **Comparison with additional DrQ baselines.** We compare SVEA implemented with DrQ [K=1,M=1] as base algorithm to DrQ with varying values of its $K, M$ hyperparameters. All methods use the *conv* augmentation (in addition to *shift* augmentation used by DrQ). Results are averaged over 5 seeds for each of the 5 tasks from DMControl-GB [21] and shaded area is ±1 std. deviation across seeds. Increasing values of $K, M$ improve sample efficiency of DrQ, but at a high computational cost; DrQ uses approx. 6x wall-time to match the sample efficiency of SVEA. *(Right)* **DistractingCS.** Episode return as a function of randomization intensity at test-time, aggregated across 5 seeds for each of the 5 tasks from DMControl-GB. See Appendix E for per-task comparison.

## 6.1 Stability and Generalization on DMControl

**Stability.** We evaluate the stability of SVEA and DrQ under 6 common data augmentations; results are shown in Figure 4. While the sample efficiency of DrQ degrades substantially for most augmentations, SVEA is relatively unaffected by the choice of data augmentation and improves sample efficiency in **27** out of **30** instances. While the sample efficiency of DrQ can be improved by increasing its K,M parameters, we find that DrQ requires approx. 6x wall-time to match the sample efficiency of SVEA; see Figure 6 *(left)*. We further ablate each component of SVEA and report both training and test curves in Figure 5; we find that both components are key to SVEA's success. Because we empirically find the *conv* augmentation to be particularly difficult to optimize, we provide additional stability experiments in Section 6.2 and 6.3 using this augmentation. See Appendix A for additional ablations.

**Generalization.** We compare the test performance of SVEA to 5 recent state-of-the-art methods for image-based RL on the `color_hard` and `video_easy` benchmarks from DMControl-GB (results in Table 1), as well as the extremely challenging DistractingCS benchmark, where camera pose, background, and colors are continually changing throughout an episode (results in Figure 6 (*right*)). We here use *conv* and *overlay* augmentations for fair comparison to SODA, and we report additional results on the `video_hard` benchmark in Appendix F. SVEA outperforms all methods considered in **12** out of **15** instances on DMControl-GB, and at a lower computational cost than CURL, PAD, and SODA that all learn auxiliary tasks. On DistractingCS, we observe that SVEA improves generalization by **42**% at low intensity, and its generalization degrades significantly slower than DrQ for high intensities. While generalization depends on the particular choice of data augmentation and test environments, this is an encouraging result considering that SVEA enables efficient policy learning with stronger augmentations than previous methods.

## 6.2 RL with Vision Transformers

Vision Transformers (ViT) [10] have recently achieved impressive results on downstream tasks in computer vision. We replace all convolutional layers from the previous experiments with a 4-layer ViT encoder that operates on raw pixels in $8 \times 8$ space-time patches, and evaluate our method using data augmentation in conjunction with ViT encoders. Importantly, we design the ViT architecture such that it roughly matches our CNN encoder in terms of learnable parameters. The ViT encoder is trained from scratch using RL, and we use the same experimental setup as in our ConvNet experiments. In particular, it is worth emphasizing that both our ViT and CNN encoders are trained using Adam [32] as optimizer and without weight decay. See Figure 7 *(top)* for an architectural overview, and refer to Appendix H for additional implementation details.

Our training and test results are shown in Figure 7 *(bottom)*. We are, to the best of our knowledge, the first to successfully solve image-based RL tasks without CNNs. We observe that DrQ overfits significantly to the training environment compared to its CNN counterpart (**94** test return on `color_hard` for DrQ with ViT vs. **569** with a ConvNet on the *Walker, walk* task). SVEA achieves comparable sample efficiency and improves generalization by **706**% and **233**% on *Walker, walk* and *Cartpole, swingup*, respectively, over DrQ, while DrQ + conv remains unstable. Interestingly, we observe that our ViT-based implementation of SVEA achieves a mean episode return of **877** on the `color_hard` benchmark of the challenging *Walker, walk* task (vs. **760** using CNNs). SVEA might therefore be a promising technique for fu-

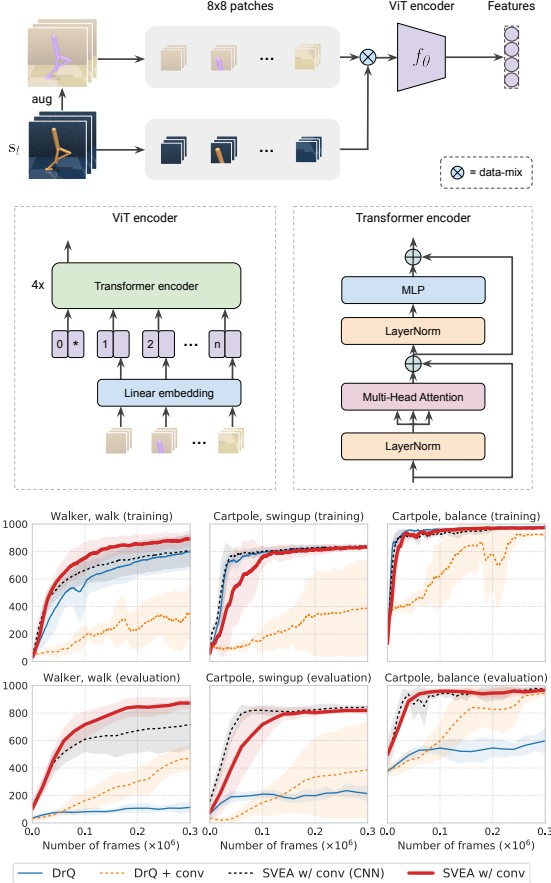

*Figure 7.* *(Top)* **ViT architecture.** Observations are divided into 144 non-overlapping space-time patches and linearly projected into tokens. Each token uses a learned positional encoding and we also use a learnable `class` token as in [10]. The ViT encoder consists of 4 stacked Transformer encoders [69]. *(Bottom)* **RL with a ViT encoder.** Training and test performance of SVEA and DrQ using ViT encoders. We report results for three tasks and test performance is evaluated on the `color_hard` benchmark of DMControl-GB. Mean of 5 seeds, shaded area is $\pm 1$ std. deviation.

ture research on RL with CNN-free architectures, where data augmentation appears to be especially important for generalization. We provide additional experiments with ViT encoders in Section 6.3 and make further comparison to ConvNet encoders in Appendix A.

*Table 2.* **Generalization in robotic manipulation.** Task success rates of SVEA and DrQ with CNN and ViT encoders in the training environment, as well as aggregated success rates across 25 different test environments with randomized camera pose, colors, lighting, and background. Mean of 5 seeds.

| Robotic manipulation | Arch. (encoder) | reach (train) | reach (test) | mv.tgt. (train) | mv.tgt. (test) | push (train) | push (test) |
|---|---|---|---|---|---|---|---|
| DrQ | CNN | **1.00** | 0.60 | **1.00** | 0.69 | **0.76** | 0.26 |
| DrQ + conv | CNN | 0.59 | 0.77 | 0.60 | 0.89 | 0.13 | 0.12 |
| **SVEA** w/ conv | CNN | **1.00** | **0.89** | **1.00** | **0.96** | 0.72 | **0.48** |
| DrQ | ViT | 0.93 | 0.14 | **1.00** | 0.16 | 0.73 | 0.05 |
| DrQ + conv | ViT | 0.26 | 0.67 | 0.48 | **0.82** | 0.08 | 0.07 |
| **SVEA** w/ conv | ViT | **0.98** | **0.71** | **1.00** | 0.81 | **0.82** | **0.17** |

## 6.3 Robotic Manipulation

We additionally consider a set of goal-conditioned robotic manipulation tasks using a simulated Kinova Gen3 arm: (i) *reach*, a task in which the robot needs to position its gripper above a goal indicated by a red mark; (ii) *reach moving target*, a task similar to (i) but where the robot needs to follow a red mark moving continuously in a zig-zag pattern at a random velocity; and (iii) *push*, a task in which the robot needs to push a cube to a red mark. The initial configuration of gripper, object, and goal is randomized, the agent uses 2D positional control, and policies are trained using dense rewards. Observations are stacks of RGB frames with no access to state information. Training and test environments are shown in Figure 8. See Appendix G for further details and environment samples.

Results are shown in Figure 9 and Figure 10. For both CNN and ViT encoders, SVEA trained with *conv* augmentation has similar sample efficiency and training performance as DrQ trained *without* augmentation, while DrQ + conv exhibits poor sample efficiency and fails to solve the *push* task. Generalization results are shown in Table 2. We find that naïve application of data augmentation has a higher success rate in test environments than DrQ, despite being less

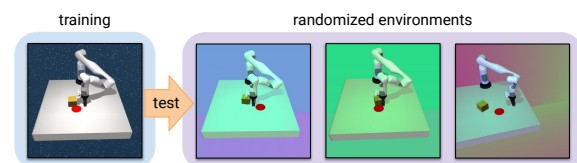

training    randomized environments    test

*Figure 8.* **Robotic manipulation.** Agents are trained in a fixed environment and evaluated on challenging environments with randomized colors, lighting, background, and camera pose.

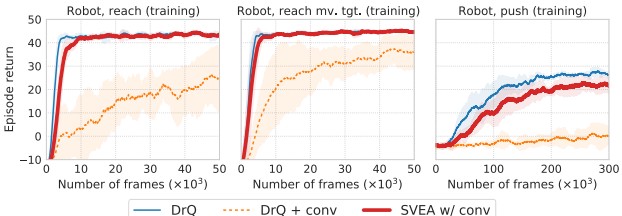

*Figure 9.* **Stability with a CNN encoder.** Training performance (episode return) of SVEA and DrQ in 3 robotic manipulation tasks. Mean and std. deviation of 5 seeds. Success rates are shown in Table 2.

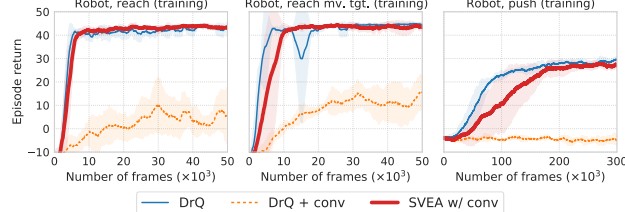

*Figure 10.* **Stability with a ViT encoder.** Training performance (episode return) of SVEA and DrQ in 3 robotic manipulation tasks. Mean and std. deviation of 5 seeds. Success rates are shown in Table 2. DrQ is especially unstable under augmentation when using a ViT encoder.

successful in the training environment, which we conjecture is because it is optimized only from augmented data. Conversely, SVEA achieves high success rates during both training and testing.

**Conclusion.** SVEA is found to greatly improve both stability and sample efficiency under augmentation, while achieving competitive generalization results. Our experiments indicate that our method scales to ViT-based architectures, and it may therefore be a promising technique for large-scale RL experiments where data augmentation is expected to play an increasingly important role.

**Broader Impact.** While our contribution aims to reduce computational cost of image-based RL, we remain concerned about the growing ecological and economical footprint of deep learning – and RL in particular – with increasingly large models such as ViT; see Appendix J for further discussion.

**Acknowledgments and Funding Transparency.** This work was supported by grants from DARPA LwLL, NSF CCF-2112665 (TILOS), NSF 1730158 CI-New: Cognitive Hardware and Software Ecosystem Community Infrastructure (CHASE-CI), NSF ACI-1541349 CC*DNI Pacific Research Platform, NSF grant IIS-1763278, NSF CCF-2112665 (TILOS), as well as gifts from Qualcomm, TuSimple and Picsart.

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
