# A Ablations

We ablate the design choices of SVEA and compare both training and test performance to DrQ and RAD. Results are shown in Table 3. We find that our proposed formulation of SVEA outperforms the test performance of all other variants, and by a large margin (method 2). Using a ViT encoder (method 1) instead of a CNN further improves both the training and test performance of SVEA, whereas the test performance of DrQ *decreases* by a factor of 5 when using a ViT encoder (method 7). This indicates that ViT encoders overfit heavily to the training environment without the strong augmentation of SVEA. We observe that both DrQ and RAD are unstable under strong augmentation (method 10 and 12). While the test performance of DrQ does *not* benefit from using a ViT encoder, we observe a slight improvement in training performance (method 7), similar to that of SVEA.

*Table 3.* **Ablations.** We vary the following choices: (i) architecture of the encoder; (ii) our proposed objective $\mathcal{L}_Q^{\text{SVEA}}$ as opposed to $\mathcal{L}_Q$ or a *mix-all* objective that uses two data-streams for both $Q$-predictions, $Q$-targets, and $\pi$; (iii) using strong augmentation (*conv*) in addition to the random shift augmentation used by default (abbreviated as *Str. aug.*); and (iv) whether the target is augmented or not (abbreviated as *Aug. tgt.*). We report mean episode return in the training and test environments (`color_hard`) of the *Walker, walk* task. Method 1 and 2 are the default formulations of SVEA using ViT and CNN encoders, respectively, method 7 and 8 are the default formulations of DrQ using Vit and CNN encoders, respectively, and method 11 is the default formulation of RAD that uses a random crop augmentation and is implemented using a CNN encoder. Mean and std. deviation. of 5 seeds.

| | Method | Arch. | Objective | Str. aug. | Aug. tgt. | Train return | Test return |
|---|---|---|---|---|---|---|---|
| 1 | **SVEA** | ViT | SVEA | ✓ | ✗ | $918_{\pm57}$ | $\mathbf{877_{\pm54}}$ |
| 2 | – | CNN | SVEA | ✓ | ✗ | $833_{\pm91}$ | $760_{\pm145}$ |
| 3 | – | CNN | SVEA | ✓ | ✓ | $872_{\pm53}$ | $605_{\pm148}$ |
| 4 | – | CNN | mix-all | ✓ | ✓ | $\mathbf{927_{\pm24}}$ | $599_{\pm214}$ |
| 5 | – | CNN | $Q$ | ✓ | ✗ | $596_{\pm55}$ | $569_{\pm139}$ |
| 6 | – | CNN | $Q$ | ✗ | ✗ | $771_{\pm317}$ | $498_{\pm196}$ |
| 7 | **DrQ** | ViT | $Q$ | ✗ | ✓ | $\mathbf{920_{\pm36}}$ | $94_{\pm18}$ |
| 8 | – | CNN | $Q$ | ✗ | ✓ | $892_{\pm65}$ | $520_{\pm91}$ |
| 9 | – | ViT | $Q$ | ✓ | ✓ | $286_{\pm225}$ | $470_{\pm67}$ |
| 10 | – | CNN | $Q$ | ✓ | ✓ | $560_{\pm158}$ | $\mathbf{569_{\pm139}}$ |
| 11 | **RAD** | CNN | $Q$ | ✗ | ✓ | $\mathbf{883_{\pm23}}$ | $400_{\pm61}$ |
| 12 | – | CNN | $Q$ | ✓ | ✓ | $260_{\pm201}$ | $246_{\pm184}$ |

# B Stability under Data Augmentation

Figure 11 compares the sample efficiency and stability of SVEA and DrQ under each of the 6 considered data augmentations for 5 tasks from DMControl. We observe that SVEA improves stability in all 27 instances where DrQ is impaired by data augmentation. Stability of DrQ under data augmentation is found to be highly sensitive to both the choice of augmentation and the particular task. For example, the *DrQ + aug* baseline is relatively unaffected by a majority of data augmentations in the *Walker, stand* task, while we observe significant instability across all data augmentations in the *Cartpole, swingup* task. Our results therefore indicate that SVEA can be a highly effective method for eliminating the need for costly trial-and-error associated with application of data augmentation.

# C Data Augmentation in RL

Application of data augmentation in image-based RL has proven highly successful [9, 36, 35, 33, 61, 21, 51] in improving generalization by regularizing the network parameterizing the $Q$-function and policy $\pi$. However, not all augmentations are equally effective. Laskin et al. [35] and Kostrikov et al. [33] find small random crops and random shifts (image translations) to greatly improve sample efficiency of image-based RL, but they empirically offer no significant improvement in generalization to other environments [21, 60]. On the other hand, augmentations such as random convolution [36] have shown great potential in improving generalization, but is simultaneously found to cause instability and poor sample efficiency [35, 21]. In this context, it is useful to distinguish between

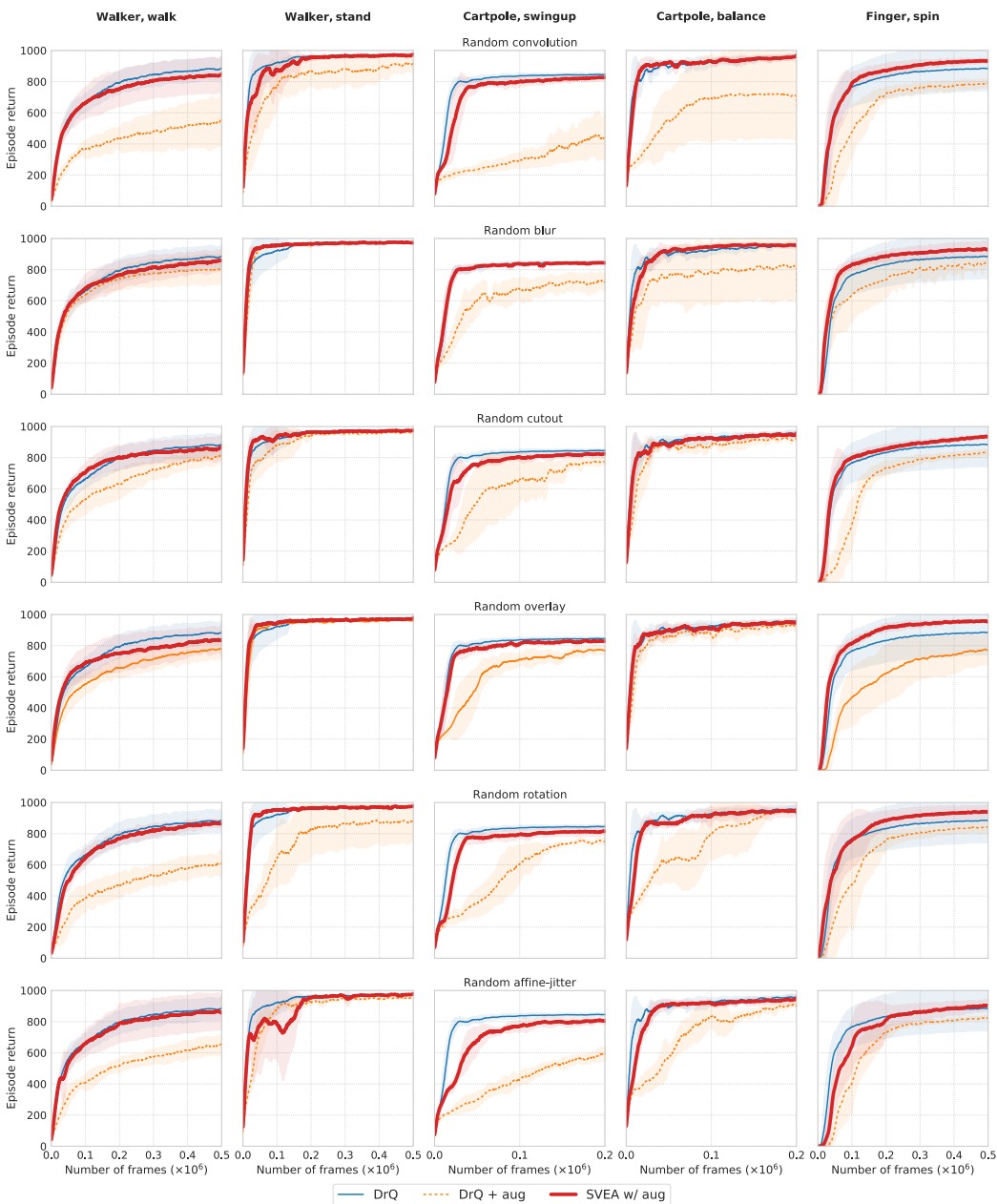

*Figure 11.* **Stability under data augmentation.** Training performance measured by episode return of SVEA and DrQ under 6 common data augmentations (using ConvNets). We additionally provide reference curves for DrQ without additional augmentation. Mean of 5 seeds, shaded area is ±1 std. deviation. SVEA obtains similar sample efficiency to DrQ without augmentation, while the sample efficiency of *DrQ + aug* is highly dependent on the task and choice of augmentation.

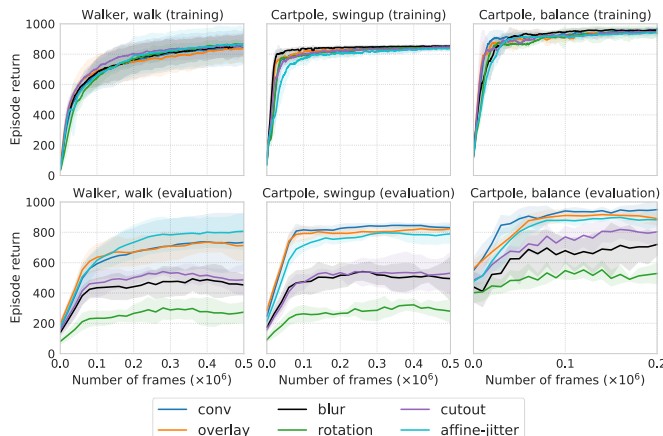

*Figure 12.* **Generalization depends on the choice of data augmentation.** A comparison of SVEA implemented using each of the 6 data augmentations considered in this work (using ConvNets). SVEA exhibits comparable stability and sample efficiency for all augmentations, but generalization ability is highly dependent on the choice of augmentation. *Top:* episode return on the training environment during training. *Bottom:* generalization measured by episode return on the `color_hard` benchmark of DMControl-GB. Mean of 5 seeds, shaded area is $\pm 1$ std. deviation.

*weak* augmentations such as small random translations that improve *sample efficiency* due to their regularization, and *strong* augmentations such as random convolution that improve *generalization* at the expense of sample efficiency. In this work, we focus on stabilizing deep $Q$-learning under strong data augmentation with the goal of improving generalization.

Figure 12 shows training and test performance of SVEA implemented using each of the 6 data augmentations considered in this work. SVEA exhibits comparable stability and sample efficiency for all augmentations, but we find that generalization ability on the `color_hard` benchmark of DMControl-GB is highly dependent on the choice of augmentation. Generally, we observe that augmentations such as *conv*, *overlay*, and *affine-jitter* achieve the best generalization, but they empirically also cause the most instability in our *DrQ + aug* baseline as shown in Figure 11.

Figure 13 provides a comprehensive set of samples for each of the data augmentations considered in this study: random *shift* [33], random convolution (denoted *conv*) [36], random *overlay* [21], random *cutout* [9], Gaussian *blur*, random *affine-jitter*, and random *rotation* [35]. We emphasize that the random convolution augmentation is not a convolution operation, but rather application of a randomly initialized convolutional layer as in the original proposal [36]. As in previous work [35, 33, 21] that applies data augmentation to image-based RL, we either clip values or apply a logistic function, whichever is more appropriate, to ensure that output values remain within the $[0, 1)$ interval that unaugmented observations are normalized to. Each of the considered data augmentations are applied to the *walker* and *cartpole* environments and are representative of the *Walker, walk*, *Walker, stand*, *Cartpole, swingup*, and *Cartpole, balance* tasks. To illustrate the diversity of augmentation parameters associated with a given transformation, we provide a total of 6 samples for each data augmentation in each of the two environments. Note that, while random shift has been shown to improve sample efficiency in previous work, it provides very subtle randomization. Stronger and more varied augmentations such as random convolution, random overlay, and affine-jitter can be expected to improve generalization to a larger set of MDPs, but naïve application of these data augmentations empirically results in optimization difficulties and poor sample efficiency.

## D    Choice of Base Algorithm

In the majority of our results, we implement SVEA using SAC as base algorithm, implemented with a random shift augmentation as proposed by DrQ [33]. We now consider an additional set of experiments where we instead implement SVEA using RAD [35] as base algorithm, which proposes to add a random cropping to SAC (in place of random shift). Training and test performances on `color_hard` are shown in Figure 15. We find SVEA to provide similar performance and benefits in terms of sample efficiency and generalization as when using DrQ as base algorithm. RAD likewise

*(a)* No augmentation (walker).                          *(b)* No augmentation (cartpole).

*(c)* Random shift (walker).                             *(d)* Random shift (cartpole).

*(e)* Random convolution (walker).                       *(f)* Random convolution (cartpole).

*(g)* Random overlay (walker).                           *(h)* Random overlay (cartpole).

*(i)* Random cutout (walker).                            *(j)* Random cutout (cartpole).

*(k)* Random blur (walker).                              *(l)* Random blur (cartpole).

*(m)* Random affine-jitter (walker).                     *(n)* Random affine-jitter (cartpole).

*(o)* Random rotation (walker).                          *(p)* Random rotation (cartpole).

*Figure 13.* **Data augmentation**. Visualizations of all data augmentations considered in this study. Left column contains samples from the *Walker, walk* and *Walker, stand* tasks, and right column contains samples from the *Cartpole, swingup* and *Cartpole, balance* tasks.

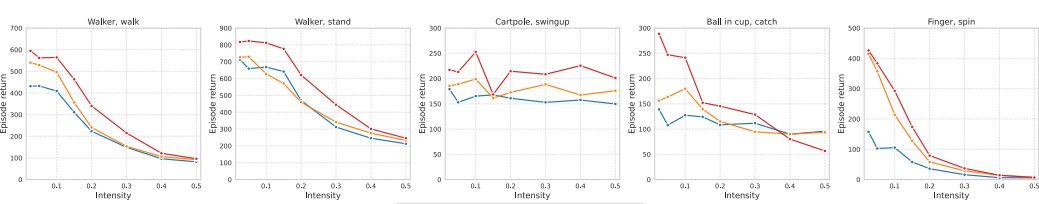

*Figure 14.* **DistractingCS.** Episode return as a function of randomization intensity, for each of the 5 tasks from DMControl-GB (using ConvNets). Mean of 5 seeds. We find that the difficulty of DistractingCS varies greatly between tasks, but SVEA consistently outperforms DrQ in terms of generalization across all intensities and tasks, except for *Ball in cup, catch* at the highest intensity.

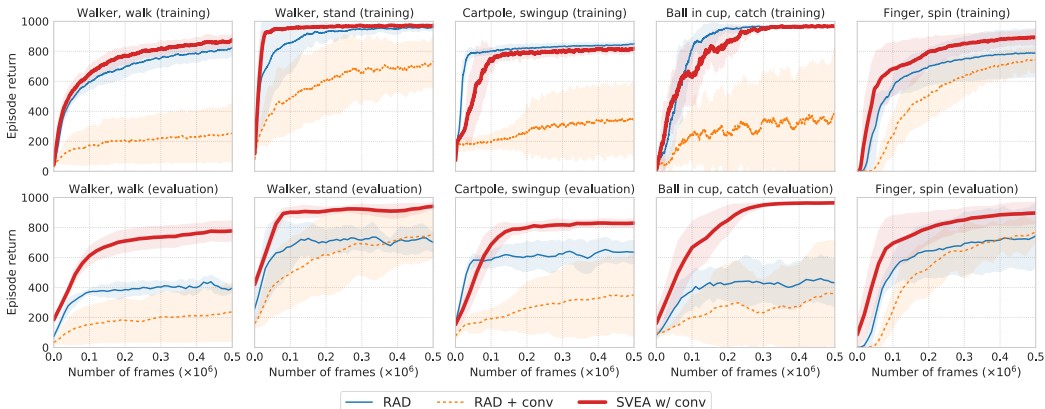

*Figure 15.* **Choice of base algorithm.** We compare SVEA implemented with RAD as base algorithm to instances of RAD with and without random convolution augmentation (using ConvNets). *Top:* episode return on the training environment during training. *Bottom:* generalization measured by episode return on the `color_hard` benchmark of DMControl-GB. Mean of 5 seeds, shaded area is $\pm 1$ std. deviation. SVEA improves generalization in all instances.

has similar performance to DrQ without use of strong augmentation, however, we observe that RAD generally is more unstable than DrQ when additionally using *conv* augmentation, and the relative improvement of SVEA is therefore comparably larger in our RAD experiments.

## E  Test Environments

Figure 16 provides visualizations for each of the two generalization benchmarks, DMControl Generalization Benchmark [21] and Distracting Control Suite [60], used in our experiments. Agents are trained in a fixed training environment with no visual variation, and are expected to generalize to novel environments of varying difficulty and factors of variation. The `color_hard`, `video_easy`, and `video_hard` benchmarks are from DMControl Generalization Benchmark, and we further provide samples from the Distracting Control Suite (DistractingCS) benchmark for intensities $I = \{0.1, 0.2, 0.5\}$. While methods are evaluated on a larger set of intensities, we here provide samples deemed representative of the intensity scale. We note that the DistractingCS benchmark has been modified to account for action repeat (frame-skip). Dynamically changing the environment at each simulation step makes the benchmark disproportionally harder for tasks that use a large action repeat, e.g. *Cartpole* tasks. Therefore, we choose to modify the DistractingCS benchmark and instead update the distractors every second simulation step, corresponding to the lowest action repeat used (2, in *Finger, spin*). This change affects both SVEA and baselines equally. Figure 14 shows generalization results on DistractingCS for each task individually. We find that the difficulty of DistractingCS varies greatly between tasks, but SVEA consistently outperforms DrQ in terms of generalization across all intensities and tasks.

## F  Additional Results on DMControl-GB

Table 1 contains results for the `color_hard` and `video_easy` generalization benchmarks from DMControl-GB. We here provide additional results for the `video_hard` benchmark, and note that we leave out the `color_easy` benchmark because it is already considered solved by previous work [21, 72]. Results are shown in Table 4. SVEA also achieves competitive performance across all 5 tasks in the `video_hard` benchmark.

## G  Robotic Manipulation Tasks

We conduct experiments with a set of three goal-conditioned robotic manipulation tasks: (i) *reach*, a task in which the robot needs to position its gripper above a goal indicated by a red mark; (ii) *reach moving target*, a task similar to (i) but where the robot needs to follow a red mark moving continuously in a zig-zag pattern at a random velocity; and (iii) *push*, a task in which the robot needs to push a cube to a red mark. We implement the tasks using MuJoCo [67] for simulation, and

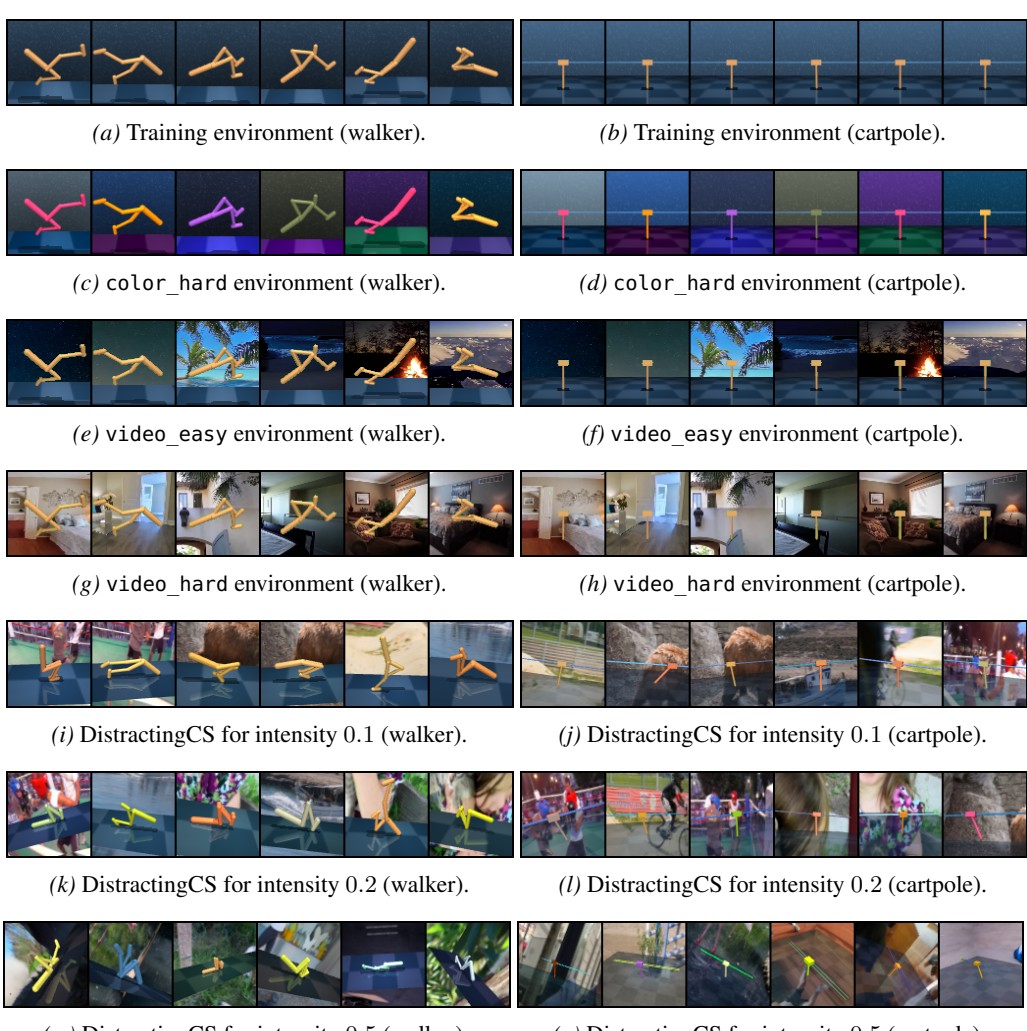

*(a)* Training environment (walker).

*(b)* Training environment (cartpole).

*(c)* `color_hard` environment (walker).

*(d)* `color_hard` environment (cartpole).

*(e)* `video_easy` environment (walker).

*(f)* `video_easy` environment (cartpole).

*(g)* `video_hard` environment (walker).

*(h)* `video_hard` environment (cartpole).

*(i)* DistractingCS for intensity 0.1 (walker).

*(j)* DistractingCS for intensity 0.1 (cartpole).

*(k)* DistractingCS for intensity 0.2 (walker).

*(l)* DistractingCS for intensity 0.2 (cartpole).

*(m)* DistractingCS for intensity 0.5 (walker).

*(n)* DistractingCS for intensity 0.5 (cartpole).

*Figure 16.* **Test environments**. Samples from each of the two generalization benchmarks, DMControl Generalization Benchmark [21] and Distracting Control Suite [60], considered in this study. In our experiments, agents are trained in a fixed training environment with no visual variation, and are expected to generalize to novel environments of varying difficulty and factors of variation.

we use a simulated Kinova Gen3 robotic arm. The initial configuration of gripper, cube, and goal is randomized, the agent uses 2D positional control, and policies are trained using dense rewards. For *reach* and *reach moving target*, at each time step there is a reward penalty proportional to the euclidean distance between the gripper and the goal, and there is a reward bonus of $+1$ when the distance is within a fixed threshold corresponding to the radius of the red mark. We use the same reward structure for the *push* task, but use the euclidean distance between the *cube* and the goal.

Each episode consists of 50 time steps, which makes 50 an upper bound on episode return, while there is no strict lower bound. All observations are stacks of three RGB frames of size $84 \times 84 \times 3$, and the agent has no access to state information. During training, the camera position is fixed, and the camera orientation follows the gripper. During testing, the camera orientation still follows the gripper, but we additionally randomize the camera position, as well as colors, lighting, and background of the environment. Samples for the robotic manipulation tasks – both in the training environment and the randomized test environments – are shown in Figure 17. We use a binary threshold for measuring task success: the episode is considered a success if the environment is in success state (i.e. either the gripper or the cube is within a fixed distance to the center of the red mark) for at least $50\%$ of all time steps in the two reaching tasks, and $25\%$ for the push task. This is to ensure that the success rate of a random policy does not get inflated by trajectories that coincidentally visit a success state for a small number of time steps, e.g. passing through the goal with the gripper.

*Table 4.* **Comparison to state-of-the-art.** Test performance (episode return) of methods trained in a fixed environment and evaluated on the `video_hard` benchmark from DMControl-GB. In this setting, the entirety of the floor and background is replaced by natural videos; see Figure 16 for samples. Results for CURL, RAD, PAD, and SODA are obtained from Hansen and Wang [21] and we report mean and std. deviation of 5 runs. We compare SVEA and SODA using the *overlay* augmentation, since this is the augmentation for which the strongest results are reported for SODA in Hansen and Wang [21]. SVEA achieves competitive results in all tasks considered, though there is still some room for improvement before the benchmark is saturated.

| DMControl-GB (`video_hard`) | CURL | RAD | DrQ | PAD | SODA (overlay) | SVEA (overlay) |
|---|---|---|---|---|---|---|
| `walker,` `walk` | 58 ±18 | 56 ±9 | 104 ±22 | 93 ±29 | **381** ±**72** | 377 ±93 |
| `walker,` `stand` | 45 ±5 | 231 ±39 | 289 ±49 | 278 ±72 | 771 ±83 | **834** ±**46** |
| `cartpole,` `swingup` | 114 ±15 | 110 ±16 | 138 ±9 | 123 ±24 | **429** ±**64** | 393 ±45 |
| `ball_in_cup,` `catch` | 115 ±33 | 97 ±29 | 92 ±23 | 66 ±61 | 327 ±100 | **403** ±**174** |
| `finger,` `spin` | 27 ±21 | 34 ±11 | 71 ±45 | 56 ±18 | 302 ±41 | **335** ±**58** |

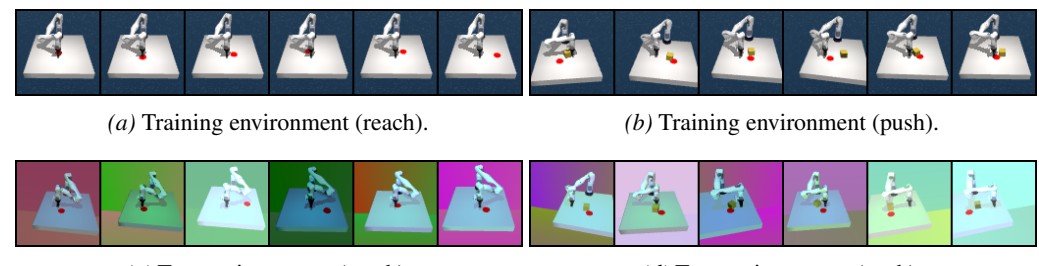

*(a)* Training environment (reach).      *(b)* Training environment (push).

*(c)* Test environments (reach).      *(d)* Test environments (push).

*Figure 17.* **Environments for robotic manipulation**. Training and test environments for our robotic manipulation experiments. Agents are trained in a fixed environment, and are expected to generalize to the unseen test environments with randomized camera pose, colors, lighting, and backgrounds.

## H   Implementation Details

In this section, we provide extensive implementation details for our experimental setup, including network architecture, hyperparameters, as well as design choices specific to our ViT experiments.

**Network architecture.** For experiments in DMControl [64], we adopt our network architecture from Hansen and Wang [21], without any changes to the architecture nor hyperparameters to ensure a fair comparison. The shared encoder $f_\theta$ is implemented as an 11-layer CNN encoder that takes a stack of RGB frames rendered at $84 \times 84 \times 3$ and outputs features of size $32 \times 21 \times 21$, where 32 is the number of channels and $21 \times 21$ are the dimensions of the spatial feature maps. All convolutional layers use 32 filters and $3 \times 3$ kernels. The first convolutional layer uses a stride of 2, while the remaining convolutional layers use a stride of 1. Following previous work on image-based RL for DMControl tasks [76, 59, 35, 33, 21], the shared encoder is followed by independent linear projections for the actor and critic of the Soft Actor-Critic [18] base algorithm used in our experiments, and the actor and critic modules each consist of three fully connected layers with hidden dimension 1024. Training takes approximately 24 hours on a single NVIDIA V100 GPU. For simplicity, we choose to apply the same experimental setup for robotic manipulation.

**Hyperparameters.** Whenever applicable, we adopt hyperparameters from Hansen and Wang [21]. We detail hyperparameters relevant to our experiments in Table 5; ViT hyperparameters are discussed in the following. We use the default SVEA loss coefficients $\alpha = 0.5, \beta = 0.5$ in all experiments using a CNN encoder.

*Table 5.* **Hyperparameters** used in experiments on DMControl and robotic manipulation.

| Hyperparameter | Value |
|---|---|
| Frame rendering | $84 \times 84 \times 3$ |
| Stacked frames | 3 |
| Random shift | Up to $\pm 4$ pixels |
| Action repeat | 2 (finger) |
| | 8 (cartpole) |
| | 4 (otherwise) |
| Discount factor $\gamma$ | 0.99 |
| Episode length | 1,000 |
| Learning algorithm | Soft Actor-Critic (SAC) |
| Number of frames | 500,000 |
| Replay buffer size | 500,000 |
| Optimizer ($\theta$) | Adam ($\beta_1 = 0.9, \beta_2 = 0.999$) |
| Optimizer ($\alpha$ of SAC) | Adam ($\beta_1 = 0.5, \beta_2 = 0.999$) |
| Learning rate ($\theta$) | 1e-3 |
| Learning rate ($\alpha$ of SAC) | 1e-4 |
| Batch size | 128 |
| SVEA coefficients | $\alpha = 0.5, \beta = 0.5$ |
| $\psi$ update frequency | 2 |
| $\psi$ momentum coefficient | 0.05 (encoder) |
| | 0.01 (critic) |

*Table 6.* **Hyperparameters** used in our ViT experiments.

| Hyperparameter | Value |
|---|---|
| Frame rendering | $96 \times 96 \times 3$ |
| Random shift | Up to $\pm 6$ pixels |
| Patch size | $8 \times 8 \times 3k$ |
| Number of patches | 144 |
| Embedding dimensionality | 128 |
| Number of layers | 4 |
| Number of attention heads | 8 |
| Score function | Scaled dot-product |
| Number of frames | 300,000 |
| Replay buffer size | 300,000 |
| Batch size | 512 |
| SVEA coefficients | $\alpha = 1, \beta = 1$ (walker) |
| | $\alpha = 1, \beta = 0.5$ (push) |
| | $\alpha = 0.5, \beta = 0.5$ (otherwise) |
| Number of ViT parameters | $489,600$ |

**RL with Vision Transformers.** We adopt a similar experimental setup for our experiments with Vision Transformers (ViT) [10] as replacement for the CNN encoder used in the rest of our experiments, but with minimal changes to accommodate the new encoder. Our ViT encoder takes as input a stack of RGB frames rendered at $96 \times 96 \times 3$ (versus $84 \times 84 \times 3$ for CNN) and uses a total of $144$ non-overlapping $8 \times 8 \times 3k$ (where $k$ is the number of frames in a frame stacked observation) image patches from a spatial grid evenly placed across the image observation. All patches are projected into 128-dimensional embeddings, and all embeddings are then forwarded as tokens for the ViT encoder. Following the original ViT implementation, we use learned positional encodings as well as a learnable `class` token. Our encoder consists of $4$ stacked Transformer [69] encoders, each using Multi-Head Attention with 8 heads. We use the ViT encoder as a drop-in replacement to CNN and optimize it jointly together with the $Q$-function using the Adam [32] optimizer and with no changes to the learning rate. We do not pre-train the parameters of the ViT encoder, and we do not use weight decay. Similar to the CNN encoder, we find ViT to benefit from random shift augmentation and we therefore apply it by default in all methods. Training takes approximately 6 days on a single NVIDIA V100 GPU. See Table 6 for an overview of hyperparameters specific to the ViT encoder experiments.

**Implementation of data augmentation in SVEA vs. previous work.** Previous work [35, 33, 61, 54] applies augmentation to both state $\mathbf{s}_t^{\text{aug}} = \tau(\mathbf{s}_t, \nu)$ and successor state $\mathbf{s}_{t+1}^{\text{aug}} = \tau(\mathbf{s}_{t+1}, \nu')$ where $\nu, \nu' \sim \mathcal{V}$. As discussed in Section C, this is empirically not found to be an issue in application of weak augmentation such as random shift [33]. However, previous work finds that strong augmentation, e.g. random convolution [36, 35], can cause instability and poor sample efficiency. We apply random

shifts in SVEA and all baselines by default, and aim to stabilize learning under strong augmentation. As such, we generically refer to observations both with and without the random shift operation as *unaugmented*, and instead refer to observations as *augmented* after application of one of the 6 augmentations considered in our study. While we provide the full SVEA algorithm in Algorithm 1, we here provide supplementary Python-like pseudo-code for the update rules of SVEA as well as generic off-policy actor-critic algorithms both with and without naïve application of data augmentation.

**Generic off-policy actor-critic algorithm.** We here provide a reference implementation for a generic base algorithm from which we will implement our proposed SVEA framework for data augmentation.

```python
def update(state, action, reward, next_state):
  """Generic off-policy actor-critic RL algorithm"""
  next_action = actor(next_state)
  q_target = reward + critic_target(next_state, next_action)
  q_prediction = critic(state, action)

  update_critic(q_prediction, q_target)
  update_actor(state)
  update_critic_target()
```

**Naïve application of data augmentation.** A natural way to apply data augmentation in off-policy RL algorithms is to augment both $\mathbf{s}_t$ and $\mathbf{s}_{t+1}$. Previous work on data augmentation in off-policy RL [35, 33, 61, 54] follow this approach. However, this is empirically found to be detrimental to sample efficiency and stability under strong data augmentation. In the following, we generically refer to application of data augmentation as an aug operation.

```python
def update_aug(state, action, reward, next_state):
  """Generic off-policy actor-critic RL algorithm that uses data augmentation"""
  state = aug(state)
  next_state = aug(next_state)

  next_action = actor(next_state)
  q_target = reward + critic_target(next_state, next_action)
  q_prediction = critic(state, action)

  update_critic(q_prediction, q_target)
  update_actor(state)
  update_critic_target()
```

**SVEA.** Our proposed method, SVEA, does *not* apply strong augmentation to $\mathbf{s}_{t+1}$ nor $\mathbf{s}_t$ when used for policy learning. SVEA jointly optimizes the $Q$-function over two data streams with augmented and unaugmented data, respectively, which can be implemented efficiently for $\alpha = \beta$ as in Algorithm 1. The following pseudo-code assumes $\alpha = 0.5, \beta = 0.5$.

```python
def update_svea(state, action, reward, next_state):
  """SVEA update for a generic off-policy actor-critic RL algorithm"""
  update_actor(state)
  next_action = actor(next_state) # [B, A]
  q_target = reward + critic_target(next_state, next_action) # [B, 1]

  svea_state = concatenate(state, aug(state), dim=0) # [2*B, C, H, W]
  svea_action = concatenate(action, action, dim=0) # [2*B, A]
  svea_q_target = concatenate(q_target, q_target, dim=0) # [2*B, 1]

  svea_q_prediction = critic(svea_state, svea_action) # [2*B, 1]

  update_critic(svea_q_prediction, svea_q_target)
  update_critic_target()
```

where $B$ is the batch size, $C$ is the number of input channels, $H$ and $W$ are the dimensions of observations, and $A$ is the dimensionality of the action space. For clarity, we omit hyperparameters such as the discount factor $\gamma$, learning rates, and update frequencies.

# I   Task Descriptions

We experiment on tasks from DMControl [64] as well as a set of robotic manipulation tasks that we implement using MuJoCo [67]. DMControl tasks are selected based on previous work on both sample efficiency [19, 76, 59, 20] and generalization [21, 60, 72], and represent a diverse and challenging skill set in the context of image-based RL. Our set of robotic manipulation tasks are designed to represent fundamental visuomotor skills that are widely applied in related work on robot learning [37, 50, 11, 44, 22, 78]. We here provide a unified overview of the tasks considered in our study and their properties; see Section G for a detailed discussion of the robotic manipulation environment. All tasks emit observations $\mathbf{o} \in \mathbb{R}^{84 \times 84 \times 3}$ that are stacked as states $\mathbf{s} \in \mathbb{R}^{84 \times 84 \times 9}$.

- *Walker, walk* ($\mathbf{a} \in \mathbb{R}^6$). A planar walker that is rewarded for walking forward at a target velocity. Dense rewards.
- *Walker, stand* ($\mathbf{a} \in \mathbb{R}^6$). A planar walker that is rewarded for standing with an upright torso at a constant minimum height. Dense rewards.
- *Cartpole, swingup* ($\mathbf{a} \in \mathbb{R}$). Swing up and balance an unactuated pole by applying forces to a cart at its base. The agent is rewarded for balancing the pole within a fixed threshold angle. Dense rewards.
- *Cartpole, balance* ($\mathbf{a} \in \mathbb{R}$). Balance an unactuated pole by applying forces to a cart at its base. The agent is rewarded for balancing the pole within a fixed threshold angle. Dense rewards.
- *Ball in cup, catch* ($\mathbf{a} \in \mathbb{R}^2$). An actuated planar receptacle is to swing and catch a ball attached by a string to its bottom. Sparse rewards.
- *Finger, spin* ($\mathbf{a} \in \mathbb{R}^2$). A manipulation problem with a planar 3 DoF finger. The task is to continually spin a free body. Sparse rewards.
- *Robot, reach* ($\mathbf{a} \in \mathbb{R}^2$). A manipulation problem with a simulated Kinova Gen3 robotic arm. The task is to move the gripper to a randomly initialized goal position. Dense rewards.
- *Robot, reach moving target* ($\mathbf{a} \in \mathbb{R}^2$). A manipulation problem with a simulated Kinova Gen3 robotic arm. The task is to continuously track a randomly initialized goal with the gripper. The goal moves in a zig-zag pattern at a random constant speed. Dense rewards.
- *Robot, push* ($\mathbf{a} \in \mathbb{R}^2$). A manipulation problem with a simulated Kinova Gen3 robotic arm. The task is to push a cube to a goal position. All positions are randomly initialized. Dense rewards.

# J   Broader Impact

The discipline of deep learning – and reinforcement learning in particular – is rapidly evolving, which can in part be attributed to better algorithms [43, 38, 18], neural network architectures [26, 69, 12], and availability of data [49, 31], but advances are also highly driven by increased computational resources and larger models such as GPT-3 [5] in natural language processing and ViT [10] in computer vision. As a result, both computational and economic requirements for training and deploying start-of-the-art models are increasing at an unprecedented rate [3]. While we are concerned by this trend, we remain excited about the possibility of reusing and re-purposing large learned models (in the context of RL: policies and value functions) that learn and generalize far beyond the scope of their training environment. A greater reuse of learned policies can ultimately decrease overall computational costs, since new models may need to be trained less frequently. As researchers, we are committed to pursue research that is to the benefit of society. We strive to enable reuse of RL policies through extensive use of data augmentation, and we firmly believe that our contribution is an important step towards that goal. Our method is empirically found to reduce the computational cost (in terms of both stability, sample efficiency, and total number of gradient steps) of training RL policies under augmentation, which is an encouraging step towards learning policies that generalize to unseen environments. By extension, this promotes policy reuse, and may therefore be a promising component both for reducing costs and for improving generalization of large-scale RL that the field appears to be trending towards.