# OpenReview forum: "Stabilizing Deep Q-Learning with ConvNets and Vision Transformers under Data Augmentation"
_NeurIPS.cc/2021/Conference — NeurIPS 2021 Poster_

### Official Review · Reviewer_UP5u · 2021-07-12

**Rating:** 4
**Confidence:** 5

**Summary:**

The paper presents a way of improving training stability of DrQ (an image-based RL algorithm that uses data augmentation) by applying image augmentation more carefully. The authors propose to input a mix of augmented and un-augmented observations into the critic, and stop augmenting the actor's inputs. On the considered benchmarks the proposed method demonstrates better generalization results.

**Ethical Concerns:**

No.

**Limitations And Societal Impact:**

Yes.

**Main Review:**

The paper identifies an issue with the increased variance of the target estimation in DrQ (image-based RL with data augmentation). The authors conjecture two root causes for the increased variance: non-deterministic Q-targets and over-regularization.
To remedy these issues they propose to apply data augmentation more carefully. Specifically, the authors suggest to mix augmented and un-augmented inputs to the critic and stop augmenting of the actor's inputs. This method is then empirically justified in a series of experiment on tasks from the DeepMind Control Suite.

Pros:
* The author investigate an interesting question of an effect of data augmentation on generalization in image-based RL.
* The method is thoroughly verified in different settings.
* The paper is clearly written and easy to follow.
* The exploration of alternative architecture, such as ViT.

Cons:
* The paper proposes a small trick that lacks in novelty. The impact of data-augmentation on the estimation variance was also studied in the original DrQ paper (Kostrikov et. al), where a very similar technique was proposed -- the target estimation was done over 2 different augmentations. This work introduces a tiny modification of the DrQ's variance reduction approach, that just combines one augmented and one un-augmented observation instead of combining two augmented observations. Furthermore, the authors fail to mention this prior result in their paper and compare to it.
* From the paper it is not clear what hyper-parameters are used for the baselines, specifically for DrQ. If the authors use the original DrQ's parameters, then I see several major issues with the empirical results: 1) SVEA uses encoder arch that consists of 11 conv layers, while DrQ only uses 4. 2) The size of replay buffer of SVEA is 500K, while DrQ's 100K. 3) SVEA virtually increases the batch size by factor of 2, which usually leads to better performance on the DeepMind Control Suite tasks, it is not clear if the authors increase batch size for DrQ or not. These discrepancies alone could result in a drastic difference in performance and can render the empirical study to be invalid.
* There is no experiments that compare individual components of SVEA to verify is that the improved performance comes from the proposed method or better/different hyper-parameters.
* The authors claim in line 30 that they "theoretically ground" their findings, but this is not the case as there is no theoretical results presented. In general, the authors don't do a good job justifying that the two pitfalls are real. Their argument is hand-wavy and is based on a series of indirect comparisons to the baseline (DrQ). Unfortunately, there are too many moving parts in those comparisons that make it virtually impossible to validate the claim (see also the point above about the hyper-parameter discrepancies).

Given the very limited scope of novelty and technical contribution, the failure to mention the prior work that is almost identical, and the inconclusive empirical study I recommend to reject this paper from a high-profile venue of NeurIPS.


**Time Spent Reviewing:**

2

---

> ### Author Response · Authors · 2021-08-10
> **Thank you and response to UP5u**
>
> We thank the reviewer for their thoughtful comments. We address individual comments in the following.
>
> **Q:** *“The paper proposes a small trick that lacks in novelty.“*
>
> **A:** We respectfully disagree that our work lacks novelty. We provide insights into the issues with using data augmentation in RL, and propose a unified framework that generalizes prior works and alleviates the identified issues, leading to substantial improvements in both sample efficiency and generalization under augmentation. We systematically study how the choice of data augmentation impacts sample eff. / generalization, and show that our proposed framework scales to RL with ViT-based encoders, which has also not been considered by prior work. Lastly, we would like to emphasize that the other reviewers appear to share our sentiment:
>
> MseA: *“The proposed framework can be seen as a generalization of existing data augmentation approaches, by considering combinations of both augmented and unaugmented data in the loss function. [...] Demonstrating that this framework still scales to ViT architectures is also promising.”*
>
> xDk5: *“While the idea of data-augmentation has been extensively used in both supervised and reinforcement learning, this paper proposes an approach that in some ways generalizes the DrQ approach, and in others simplifies it. In my view the modifications, though subtle, are novel.”*
>
> 7gSX: *“The paper reviews previous attempts to learn robust state representations to visual variations and formulates clear hypotheses for explaining the causes of difficulties which prior methods faced and were hindered by. The said hypotheses suggest a simplified approach (SVEA) which is empirically developed and evaluated in the remainder of the paper.”*
>
> ----
>
> **Q:** *“The impact of data-augmentation on the estimation variance was also studied in the original DrQ paper (Kostrikov et. al), where a very similar technique was proposed -- the target estimation was done over 2 different augmentations. This work introduces a tiny modification of the DrQ's variance reduction approach that just combines one augmented and one un-augmented observation instead of combining two augmented observations. Furthermore, the authors fail to mention this prior result in their paper and compare to it.”*
>
> **A:** Thank you for the feedback. We would like to stress that while DrQ reduces variance by averaging over different augmentations, SVEA does not augment observations used in Q-targets, which drastically reduces variance when training with strong augmentation. We will make sure that it is made more clear in the final version of the paper. We will provide additional comparisons to DrQ with varying K,M parameters but note that it adds substantial computational overhead, approx. 6x wall-time for K=4,M=4 -- see our general comment on this.
>
> ----
>
> **Q:** *“From the paper it is not clear what hyper-parameters are used for the baselines, specifically for DrQ. If the authors use the original DrQ's parameters, then I see several major issues with the empirical results: 1) SVEA uses an encoder arch that consists of 11 conv layers, while DrQ only uses 4. 2) The size of replay buffer of SVEA is 500K, while DrQ's 100K. 3) SVEA virtually increases the batch size by factor of 2, which usually leads to better performance on the DeepMind Control Suite tasks, it is not clear if the authors increase batch size for DrQ or not. These discrepancies alone could result in a drastic difference in performance and can render the empirical study to be invalid.”*
>
> **A:** We appreciate the reviewer’s concerns. As stated in L248-253, we implement our method and all baselines using SAC as the base algorithm, and we use the **same network architecture and hyperparameters for all methods**. To be clear, this means that all variants of both SVEA and DrQ, as well as the RAD, CURL, PAD, SODA baselines use 11 conv layers and a replay buffer of size 500k. Our experimental setup is adopted from [17,18,61] without any modifications and we provide further details and hyperparameters in Section 8 of the supplementary material. Lastly, we want to emphasize that SVEA only increases batch size in Q-predictions, and that the added samples are in fact **duplicates**, i.e. the loss and gradient would be identical without use of data augmentation (e.g. alpha=1, beta=0 or simply alpha=0.5, beta=0.5 with tau being the identity function). We agree with the sentiment though and will provide additional DrQ baselines in the final version of the paper; see our general comment for preliminary results. We welcome further discussion on this topic and are open to provide other comparisons as the reviewers see fit.
>
> ----
>
> **Q:** *“There is no experiments that compare individual components of SVEA to verify is that the improved performance comes from the proposed method or better/different hyper-parameters."*
>
> **A:** Figure 4 in the manuscript and Section 1 (Ablations) in supplementary material ablate individual components of SVEA, and all methods use the same hyperparameters in our experiments; see above comment.
>
> ----
>
> **Q:** *“The authors claim in line 30 that they "theoretically ground" their findings, but this is not the case as there is no theoretical results presented.”*
>
> **A:** We agree with this sentiment and will rephrase the sentence.

---

> ### Author Response · Authors · 2021-09-02
> **Please let us know whether you have further questions**
>
> Dear reviewer,
>
> We kindly remind you that we have provided additional results and clarifications based on your review. Since the discussion period is coming to an end, we would highly appreciate if the reviewer could go over our responses and let us know if you have any additional concerns or questions.
>
> Thank you,

---

### Official Review · Reviewer_7gSX · 2021-07-20

**Rating:** 6
**Confidence:** 5

**Summary:**

The paper addresses the question of zero-shot or fast generalization of value-based deep reinforcement learning agents using visual inputs to image transformation which do not change the (conceptual) state of the environment. The long term goal is therefore training agents which learn efficient, high return control policies both which are robust to irrelevant variations in high dimensional input observations. The ultimate goal seems to be generalization of learned skills to novel environments, but as far as I can tell this particular evaluation philosophy is implemented in experiments.

The paper reviews previous attempts to learn robust state representations to visual variations and formulates clear hypotheses for explaining the causes of difficulties which prior methods faced and were hindered by. The said hypotheses suggest a simplified approach (SVEA) which is empirically developed and evaluated in the remainder of the paper. Control tasks from widely used benchmarks are used to evaluate the method, somewhat less usual is the focus on vision-based high-dimensional versions of such tasks, instead of standard domains such as Atari games and benchmarks for generalization such as Natural Atari [1], CoinRun/ProcGen [2].



**Limitations And Societal Impact:**

I understand why every paper claims the maximal general applicability possible which cannot be refuted by reviewers without experiments; please note that proving negatives is hard, and much rests on your honesty and expertise alone. I am not comfortable with text which talks about aspirational goals as if they were achieved in this paper. The abstract talks about generalization to novel environments and great improvements in stability and sample efficiency, it seems to me that not enough attention to detail has been expended in quantifying the limitations of this approach and the wider family of algorithms to which it belongs. For example, a claim of "greatly improved sample efficiency" sounds outstanding, but then it is not clear to a non-RL specialist how many order of magnitude we are talking about, and how augmentations themselves make the problem worse. While the paper deserves credit for exploring this subject in detail, such claims are vague at best and may not age well; sample efficiency may mean very different amounts of experience in 5 years time.

**Main Review:**

I am generally happy with the paper, I have a few suggestions:

* The goal of reinforcement learning is discovering an optimal control policy, not necessarily an action-value function. Strictly speaking, value estimation is just a crutch. Hence, why bother with robust value estimation? Why not learn value functions without augmentations, use them to discover an optimal policy, and then use supervised learning [3, 4] to encode a robust policy? Please clarify in the text! The proposed method is attempting to solve a problem (RL under observation augmentation) which may not be relevant if all that is needed is good control under diverse observation augmentations. How am I getting this wrong?
* Indeed, for all these environments it is not clear that value prediction from vision is needed at all, since a policy could be derived much faster from full state representations in terms of sample efficiency and computational speed, and then have that policy supervise a vision to action mapping, circumventing the robust value estimation problem altogether.
* There are many possible definitions of generalization in reinforcement learning; one is accurate value prediction in unseen states, as realistic problems have enormous state spaces; another is robust value prediction from high-dimensional observations, where states may have been seem before but there is extraneous variance in the observations which needs to be dealt with. Finally, there are progressively stronger definitions of generalization, namely across domain gaps which may or may not introduce variability relevant to the control policy itself. Which definition have you tested for in your experiments?
* Could you please clarify in the main text which experiments use some form of state information, e.g. "proprioceptive" data common in control tasks, as well as target object information, and which do not? If such information is used, how does it reflect on the ability to extrapolate the presented results to problems where such information is not available.
* Could you please specify for each set of results whether the set of initial states was randomized and by how much? One surefire way to get robustness to high variability in observations is open-loop control, which can also be implemented by non-recurrent architectures, e.g. by relying heavily on proprioceptive inputs or general statistics of observations.


References:
1. (Natural Atari) Amy Zhang et al. 2018. Natural Environment Benchmarks for Reinforcement Learning. https://arxiv.org/abs/1811.06032
2. (CoinRun/ProcGen) Karl Cobbe et al. 2019. Leveraging Procedural Generation to Benchmark Reinforcement Learning. https://arxiv.org/abs/1912.01588
3. Andrei Rusu et al. 2015. Policy Distillation. https://arxiv.org/abs/1511.06295
4. Wojciech Czarnecki et al. 2019. Distilling Policy Distillation. https://arxiv.org/abs/1902.02186


**Time Spent Reviewing:**

3

---

> ### Author Response · Authors · 2021-08-10
> **Thank you and response to 7gSX**
>
> We thank the reviewer for their thoughtful comments. We address individual comments in the following.
>
> **Q:** *“[...] Why not learn value functions without augmentations, use them to discover an optimal policy, and then use supervised learning [3, 4] to encode a robust policy? [...]”*
>
> **A:** Although not the main focus of our work, we emphasize that previous works on data augmentation in RL have found that data augmentation can not only improve generalization, but also sample efficiency (e.g. RAD [28] and DrQ [26]). However, these works also show that depending on the particular task and choice of data augmentation, training performance may vary significantly. Our work reduces the brittleness of Q-learning algorithms under data augmentation, which ultimately reduces the need for experimentation, and we observe that strong augmentations improve both training and test performance in some instances. See e.g. Figure 6 (right) of our manuscript and Section 2 of supplementary material, where we find our approach also achieves better sample efficiency in training envs using augmentations. We will make this more clear in the final version of the paper.
>
> ----
>
> **Q:** *“[...] a policy could be derived much faster from full state representations in terms of sample efficiency and computational speed, and then have that policy supervise a vision to action mapping [...]”*
>
> **A:** This is in principle a great approach, and is in fact similar in spirit to concurrent work* on generalization in RL. However, there are many instances in which full state information may not be available, and our method is therefore more generally applicable. Further, imitation learning strategies typically require a large number of trajectories, which can make them impractical to use. We welcome future work that addresses each of these challenges.
>
>
> \* Fan et al., Self-Expert Cloning for Zero-Shot Generalization of Visual Policies, 2021
>
> ----
>
> **Q:** *“There are many possible definitions of generalization in reinforcement learning [...] Which definition have you tested for in your experiments?”*
>
> **A:** We investigate generalization to high-dimensional observations from unseen environments, e.g. visual (and potentially non-stationary) distribution shifts such as colors, video backgrounds, and camera poses, as shown in Figure 3+7. See Section 3 (problem formulation) for a definition of our problem setting and assumptions, and L140 for its implications on generalization.
>
> ----
>
> **Q:** *“Could you please clarify in the main text which experiments use some form of state information, e.g. "proprioceptive" data common in control tasks, as well as target object information, and which do not?”*
>
> **A:** As described in L251 and L336 observations are stacks of RGB frames with no access to state information, which is arguably more general. In robotic manipulation tasks, the goal is indicated by a red mark in the environment itself, which is provided via visual input rather than state information (see Figure 7), which would be straightforward to implement in a real world setup.
>
> ----
>
> **Q:** *“Could you please specify for each set of results whether the set of initial states was randomized and by how much?”*
>
> **A:** Initial states are randomized in all tasks considered (to prevent rote memorization). In DMControl, we use the default settings proposed in the original benchmark [55]. In robotic manipulation tasks, we randomize the initial configuration of robot, object, and goal, as described in e.g. L332; see Section 7+9 in supplementary material for initial state samples and further task descriptions.
>
> ----
>
> **Q:** *“The abstract talks about generalization to novel environments and great improvements in stability and sample efficiency, it seems to me that not enough attention to detail has been expended in quantifying the limitations of this approach and the wider family of algorithms to which it belongs.”*
>
> **A:** We appreciate the feedback and will adjust the final version to make this more clear. Our work can be considered a generalization of previous work on data augmentation and therefore largely have the same limitations aside from the stability issues that our method addresses. As described in Section 3 (problem formulation), we only consider generalization in observations; generalization to e.g. novel dynamics using data augmentation remains a largely unexplored topic in the literature and it is unclear whether it can even be made compatible with model-free methods without relying on extensive domain knowledge.

---

> > ### Comment · Reviewer_7gSX · 2021-08-26
> > **Rebuttal acknowledgement**
> >
> > I thank the authors for the meticulous rebuttals and the reviewers for their thoughtful input!
> >
> > As for my concerns, I will discuss all of them in order since I do not any believe they are fully addressed yet:
> > 1. The answer doesn't fully address this question; one way to achieve the same result (high return behavior under data augmentations) is not to use deep Q-learning with data augmentation at all, but rather learn in whatever way works best on the original task without augmentations and then encode the "optimal" policy in a supervised fashion. While this may not be the main focus of the work, it is surely an alternative way to achieve the same result, and may even be more sample efficient. Please keep in mind that you do not report how accurate your Q-predictions are, but how good the resulting behavior is. Hence, it is up to you to prove that you're solving a problem that has no way around it if you want to claim significance beyond stabilizing Q-learning under augmentations, which may or may not matter for learning robust behaviors. Such proof should increase the significance of the results if done correctly.
> > 1. Please note that imitation learning and behavior cloning/policy distillation are not the same paradigm, there are considerable conceptual and practical differences. While imitation learning may be more sample inefficient, supervised re-encoding of policies can be much more sample efficient than RL, especially under heavy data augmentation or complex joint objectives, see papers cited in the review. While the proposed approach may be more general, it is demonstrated *only* in environments where state information is readily available, and *not* in one case where it is not. Hence, the claims of being strictly more general are purely conceptual, and I could not find an experiment in the paper to argue for this claim. Example of environments where state information is not readily available are those based on e.g. Atari, ProcGen or CoinRun, etc.
> > 1. I still think that the text should clarify that improved generalization doesn't mean that generalization to the original, non-augmented environment observation space is better, but rather generalization across augmentations is improved. Furthermore, for each experiment it should be clarified which augmentations were used for training, including their parameter ranges, and which were used for testing. The remaining gap should be made abundantly clear in order to give a complete definition of generalization.
> > 1. Acknowledged.
> > 1. Acknowledged, but significance and novelty should not be inflated.

---

> > > ### Author Response · Authors · 2021-08-29
> > > **Thank you and response**
> > >
> > > We thank the reviewer for their continued discussion and thoughtful inputs. Your time and effort is highly appreciated, and we therefore attempt to address your remaining concerns as succinctly as possible.
> > >
> > > **Q:** *“[...] one way to achieve the same result (high return behavior under data augmentations) is not to use deep Q-learning with data augmentation at all, but rather learn in whatever way works best on the original task without augmentations and then encode the "optimal" policy in a supervised fashion. While this may not be the main focus of the work, it is surely an alternative way to achieve the same result, and may even be more sample efficient. [...]”*
> > >
> > > **A:** We agree that this is a very reasonable approach. We would like to stress that data augmentation has been established as a **tremendously effective** component for improving RL from images **during both training and testing**. In fact, most DMControl tasks have **not** been solved from images with model-free RL methods **without** use of data augmentation. We exclude the SAC baseline without augmentation in our paper as it fails to solve any tasks to a satisfactory level, let alone generalize to unseen environments. For supporting evidence, see e.g. RAD [28] Table 1 where data augmentation increases training performance (return) of SAC from 42 to 918 (on a scale from 0 to 1000 return) at 500k steps for the *walker, walk* task of DMControl, and DrQ [26] Figure 1 shows similar gains across a variety of DMControl tasks and network architectures. It is therefore highly valuable to also apply data augmentation during policy learning, but discovering the “right” choice of data augmentation (and intensity) for a particular task -- i.e. balancing both training and test benefits from augmentation and its implications on optimizability -- is not trivial and requires experimentation, as previously explored in [18, 26, 28, 42]. See L71-81 of our manuscript or Section 4 of the supplementary material for an extended discussion on this matter. In this context, our work offers intuition for why data augmentation makes optimization more difficult, and provides a simple method for alleviating the issues. We highly appreciate this discussion with the reviewer and will update the paper to make it more clear why data augmentation is needed beyond generalization.
> > >
> > > ----
> > >
> > > **Q:** *“Please note that imitation learning and behavior cloning/policy distillation are not the same paradigm, there are considerable conceptual and practical differences. [...] While the proposed approach may be more general, it is demonstrated only in environments where state information is readily available, and not in one case where it is not. [...]”*
> > >
> > > **A:** We acknowledge this distinction and we appreciate that the reviewer agrees that our approach is more general. We choose DMControl because it is a well established benchmark for image-based RL, and particularly so for recent methods on representation learning and data augmentation in RL. In fact, the availability of results for related work on this benchmark makes it easier for readers to gauge the significance of our method, which we strongly value. In our view, existence of a state-based DMControl benchmark does not detract from the relevance of literature using image-based DMControl any more than e.g. the RAM version of Atari does for image-based Atari; they are simply separate problem settings. That aside, we built our robotic manipulation tasks in MuJoCo (not a part of DMControl). Learning manipulation directly from pixels in this case is particularly useful since the estimation of object location/orientation/shape from an image is challenging. In general, we believe that there is great value in algorithms capable of learning generalizable policies directly from images (which the field also appears to trend towards), but we acknowledge the legitimacy of the reviewer’s proposal to use state information + policy distillation and encourage research in this direction.
> > >
> > > ----
> > >
> > > **Q:** *“I still think that the text should clarify that improved generalization doesn't mean that generalization to the original, non-augmented environment observation space is better, but rather generalization across augmentations is improved. Furthermore, for each experiment it should be clarified which augmentations were used for training, including their parameter ranges, and which were used for testing.”*
> > >
> > > **A:** We agree with the sentiment and will clarify our problem setting further in the main text such that there can be no misunderstanding of what we mean by generalization. We already provide extensive details on our problem setting in the supplementary material, though we agree that this should be absolutely clear from the main text alone. Section 4 and 5 of the supplementary material discusses data augmentation and generalization to other environments, and we further provide 12 samples for each choice of data augmentation in Figure 4 of the supplementary material, as well as 12 samples from each test environment distribution considered in Figure 6 of the supplementary material. The DMControl-GB test environments were proposed by [11] and the DistractingCS test environments were proposed by [27], both of which are open source and used in related work. For DMControl-GB which offers a discrete set of distribution shifts and difficulties, we report results on each of the 3 hardest settings. For DistractingCS which offers a continuous intensity parameter in [0,1], we report results across 8 intensities in the [0,0.5] range as indicated in Figure 6 of our manuscript. Policies are trained only in the default DMControl environments with no visual variation and we rely solely on data augmentation for generalization to the (unseen) test environments. In robotic manipulation tasks, we randomize the following simulation parameters during testing:
> > >
> > > * Field of view (45 to 50, default is 45)
> > > * Camera position (Gaussian distribution with std=0.015m)
> > > * Brightness scaling factor (Gaussian distribution with std=0.2, default is 1)
> > > * Lighting, diffuse (uniform distribution, [0,0.35] for each color channel, default is 0)
> > > * Background, color gradient (uniform distribution, full range of values)
> > >
> > > We do not randomize simulation parameters during training. Whenever possible, we use the same implementation and parameter ranges for data augmentation as in related work. We apologize if we have misinterpreted the reviewer’s response or if they believe that the experimental setup has not been made sufficiently clear in the manuscript. We will provide the above list of simulation parameters for the robotic manipulation tasks in the final version of the paper, as well as parameter ranges and further implementation details for each of the considered data augmentations. Our code will be open sourced which should also help in this regard.
> > >
> > > We again thank the reviewer for their feedback and continued discussion.

---

### Official Review · Reviewer_xDk5 · 2021-07-25

**Rating:** 7
**Confidence:** 4

**Summary:**

This work proposes a simple technique to improve the robustness and sample-efficiency of reinforcement learning. The technique can be applied to any algorithm via a simple modification of the input pipeline to said algorithm's Q-learning subroutine. The claim is that this modification reduces the variance of Q-targets and therefore stabilizes learning.

**Limitations And Societal Impact:**

I did not see any mention of limitations. The authors mentioned the growing footprint of large models and how this work could help negotiate that.

**Main Review:**

Disclaimer: Emergency review with all the short-comings that comes with.

# Originality

While the idea of data-augmentation has been extensively used in both supervised and reinforcement learning, this paper proposes an approach that in some ways generalizes the DrQ approach, and in others simplifies it. In my view the modifications, though subtle, are novel.

# Quality

The work produces an extensive array of empirical results showing reobust generalization and impressive sample-efficiency. The comparisons to DrQ lack specification of the hyperparamters K and M, which can easily be added. If, as I suspect, the comparison is with DrQ[K=1, M=1], it would be nice to see a comparison to the default DrQ[K=2, M=2] and/or at the very least DrQ[K=2, M=1], which matches the number of forward passes through the Q-network.

Similarly, in the empirical demonstration of Pitfall 2 (Figure 1) it would be nice to see the equivalent plot for various values of K.

Finally, I would slightly quibble with the claim of "theoretically grounded" on line 30, as this paper does not feature any rigorous theoretical statements.

# Clarity

Overall the paper is very clearly and well written. The only major point of contention is that DrQ is parameterized by a K and M which control how many samples the target Q-values and TD loss, respectively, are averaged over. It would be good to specify which is being used here. It sounds like [K=1, M=1] (?) but I would think the fairest comparison would be [K=2, M=1].


## Minor points

- The targets on line 215 and line 9 of Algorithm 1 presumably look different in continuous control cases, which is the main focus of the experimental section.
- Very minor point: would the expressions not be equivalent but simpler if the trade-off was simply alpha and (1 - alpha), instead of beta?
- The experimental setup is slightly unclear in the text, copying the caption of Figure 3 into the main body would help. In particular, the sentence "all methods are trained for 500k frames and evaluated on the full set of tasks".
- Figure 6 (left) needs error-bars.

# Significance

Data-augmentation seems to be a very simple technique for improving the sample-efficiency and generalizability of RL agents. This is particularly important for real-world applications where interactions with the environment can be expensive and the sim-to-real gap can be prohibitive.

**Time Spent Reviewing:**

4

---

> ### Author Response · Authors · 2021-08-10
> **Thank you and response to xDk5**
>
> We thank the reviewer for their thoughtful comments. We address individual comments in the following.
>
> **Q:** *“The comparisons to DrQ lack specification of the hyperparamters K and M, which can easily be added”*
>
> **A:** For fair comparison, we implement DrQ using K=1, M=1 and implement our changes directly on top of the DrQ baseline with no changes to hyperparameters nor architecture. We note that larger values of K,M for DrQ add significant computational overhead compared to SVEA; see our general comment on this. We will clarify our choice of K,M for the DrQ baselines and provide a comparison with varying K,M for DrQ in the final version.
>
> ----
>
> **Q:** *“Similarly, in the empirical demonstration of Pitfall 2 (Figure 1) it would be nice to see the equivalent plot for various values of K.”*
>
> **A:** We will add this to the appendix of the final version.
>
> ----
>
> **Q:** *“I would slightly quibble with the claim of "theoretically grounded" on line 30, as this paper does not feature any rigorous theoretical statements.”*
>
> **A:** The authors agree with this sentiment and will rephrase the sentence.
>
> ----
>
> **Q:** *“The targets on line 215 and line 9 of Algorithm 1 presumably look different in continuous control cases, which is the main focus of the experimental section.”*
>
> **A:** Correct; in the continuous case, max is approximated using a neural network, e.g. in DDPG, SAC, RAD, DrQ, SVEA. We abstracted away this detail for brevity but will clarify the distinction in the final version.
>
> ----
>
> **Q:** *“The experimental setup is slightly unclear in the text, copying the caption of Figure 3 into the main body would help. In particular, the sentence "all methods are trained for 500k frames and evaluated on the full set of tasks".”*
>
> **A:** We appreciate the feedback and will make this more clear. To be clear, we follow the experimental setup of [17,18,61] and use the same hyperparameters and architecture for all methods considered, and we evaluate methods in both the training environment and a variety of test environments proposed by [18] and [51], as well as in robotic manipulation tasks with simulated distribution shifts.

---

### Official Review · Reviewer_MseA · 2021-07-30

**Rating:** 6
**Confidence:** 4

**Summary:**

This paper studies the problem of high-variance Q-targets under data augmentation, and proposes a method to reduce the variance of the targets called SVEA. The key idea contains three parts: (1) Apply augmentation to the Q-value estimation of the current state and not the next state Q-targets for bootstrapping, and (2) modify the objective to include both the targets using augmented and original observations and (3) optimize the actor using only the unaugmented data for actor-critic algorithms, with parameter sharing.

The paper performed experiments in DeepMind control suite and its variants (with raw pixel inputs). Compared to other data augmentation methods (e.g. DrQ) their proposed SVEA method is robust to the type of data augmentation and is more sample efficient with a better final performance. Some ablations were also presented among the above components to indicate the usefulness of all components. Generalization performance was demonstrated in the DistractingCS and robotic manipulation environments. The method also appears to help when using a Vision Transformers architecture.


**Ethical Concerns:**

No ethical issues with this paper.

**Limitations And Societal Impact:**

Yes, the authors have adequately addressed the limitations and potential negative societal impact of their work.

**Main Review:**

**Originality**: The proposed framework can be seen as a generalization of existing data augmentation approaches, by considering combinations of both augmented and unaugmented data in the loss function. The related work section is adequately touching upon the several areas which this framework can be viewed from.

**Quality**: There is no theoretical analysis for the proposed framework, but the paper presents several empirical results to back up the claims, including ablations on the various components. However, I would suggest backing up the claim on Pitfall 1: Non-deterministic Q-target with some empirical data as well. How bad is the variance with non-deterministic Q-targets in practice? With SVEA, even though we have deterministic Q-targets, we still have noisy predictions due to the data augmentation. So there can still be comparison of variance in the case of combination of augmented and unaugmented current state and next state.

For the alpha and beta hyperparameter: it was mentioned that 0.5/0.5 works well for the experiments. Have the authors tried varying this parameter throughout the training (i.e. some sort of annealing schedule), and whether that hinders or helps the performance even more?

**Clarity**: The paper is overall fairly clear to read.

**Significance**: Given the recent popularity of applying data augmentation in vision-based reinforcement learning agents, I believe that the proposed technique, which is simple to implement, can be easily incorporated as a toolbox of RL practitioners/researchers. Demonstrating that this framework still scales to ViT architectures is also promising. It would be nice to have some more theoretical justification for the framework by the original authors, but perhaps that could be a follow up work (potentially by others).


**Time Spent Reviewing:**

7

---

> ### Author Response · Authors · 2021-08-10
> **Thank you and response to MseA**
>
> We thank the reviewer for their thoughtful comments. We address individual comments in the following.
>
> **Q**: *“I would suggest backing up the claim on Pitfall 1: Non-deterministic Q-target with some empirical data as well. How bad is the variance with non-deterministic Q-targets in practice?”*
>
> **A:** Thank you for your suggestion. We are running this experiment in response to your comment and provide preliminary results here*. We evaluate Q-target variance of SVEA and the DrQ + conv baseline in three tasks from DMControl and across 3 seeds, i.e. curve is mean variance and shaded area is +-1 std. deviation across seeds. Variance is continuously estimated from multiple forward passes with the target Q-function using the same observation. For DrQ + conv, this amounts to Q-value estimations on the same observations using different augmentations; for SVEA, no (strong) augmentation is applied to the target which dramatically reduces variance, but some variance remains due to action sampling from the policy. We observe a 640x reduction in variance in the cartpole, swingup task, and 18x and 31x reduction in the walker, walk and walker, stand tasks, respectively (measured at 300k steps). We will add a more rigorous evaluation of Q-target variance in the final version of the paper.
>
> \* https://drive.google.com/file/d/1_QCA7FxBDkemqKQVWi3e_wFmVBfox64m/view?usp=sharing
>
> ----
>
> **Q:** *“For the alpha and beta hyperparameter: it was mentioned that 0.5/0.5 works well for the experiments. Have the authors tried varying this parameter throughout the training (i.e. some sort of annealing schedule) [...]"*
>
> **A:** This is an interesting proposal. We generally do not find our algorithm to be very sensitive to alpha/beta hyperparameters, but we imagine that a gradual increase in augmented data as training progresses could indeed be helpful in cases where optimization is particularly challenging. This would however introduce additional hyperparameters that may not be as trivial to choose, and would also detract from any benefits in sample efficiency that data augmentation may provide during training. For simplicity, we therefore recommend constant 0.5/0.5 values for the vast majority of use cases. We will add further discussion on this in the final version of the paper.

---

> > ### Comment · Reviewer_MseA · 2021-08-23
> > **Thank you for the clarifications**
> >
> > I have read the author's rebuttal comments as well as the other reviewer's concerns. It is great that the explicit variance comparisons between SVEA and DrQ clearly demonstrates the variance reduction with the proposed method. I will still keep my original score in the review, but I want to convey that the additional experiments helped.

---

### Official Review · Reviewer_xUWy · 2021-09-03

**Rating:** 6
**Confidence:** 4

**Summary:**

(Auxiliary Review only -- the authors need not respond to this).

This work proposes issues with data augmentation in RL and proposes a couple of techniques to address those issues. It applies augmentation only on the current state and not future state to address erroneous bootstrapping. It also proposes a Q objective that uses both the augmented and un-augmented data. Also, for actor critic experiments, the actor is optimised only with un-augmented data and learn a  policy that is general via sharing parameters between the networks. These are simple techniques that help the paper demonstrate good results in a range of tasks.

This is a clear paper with good set of experiments. The problem has been described clearly and the work proposes simple refinements to the data augmentation strategies used in RL. While I am convinced of the quality of the method itself and the experiments, this work could have benefitted from more rigour with the baselines to make a stronger case for the method. Hence, I am offering a score of 6 noting that the paper is marginally above acceptance threshold.

**Ethical Concerns:**

There are no ethical concerns that I am aware of.

**Ethics Review Area:**

["I don’t know"]

**Limitations And Societal Impact:**

Yes, the authors have addressed the limitations.

**Main Review:**

Note : * This is an AUXILIARY REVIEW only. The authors are not expected to respond to this review or any of the concerns presented here. * *

The paper is well-written, clear and provides a good description of the experiments. It clearly sets up the problem that standard data augmentation methods suffer from. It also provides empirical solutions to these problems and backs up these claims with experimental results. Even though the paper justifies the claims only empirically based on experimentation, the ideas proposed here have made simple and easily extendable modifications, which can be very useful in complex state action spaces when RL+Data augmentation is employed. The ideas proposed here are pretty simple and straightforward, but I am not convinced these methods alone provide us a full-fledged framework for data augmentation as claimed. Also, any theoretical justification provided with this work is not rigorous enough to demonstrate that these methods are universal for all data augmentation problems.  On the novelty front, the paper's key contribution seems to be the idea of using un-augmented inputs to the targets. There are two ways to look at this -- just as a simple innovation being not novel enough on its own or trying to understand how the work arrives at this simple innovation and whether the innovation helps improve the algorithm performance over a wide variety of tasks. I am more inclined towards the latter. I believe this is still a useful enough contribution to practitioners and researchers alike, where a few simple modifications help greatly stabilize performance.

My major concern with the paper was the hyper-parameters and architectures used for the baselines. It looks like the adapted baselines were handicapped a bit by architectural changes and these baselines were not tuned to optimality, or at least it was not very evident that this was done thoroughly. This takes a bit of the shine off the impressive results presented by the paper and I am not sure if tuning the parameters would change any relative comparison between those baselines and this method.

The paper's proposed methods are good and the results are useful on their own to the community. Having said that, because of the other minor issues mentioned earlier, I am not able to give a higher score than 6 for this paper.

**Time Spent Reviewing:**

4 hours

---

### Author Response · Authors · 2021-08-10
**Thank you and general comments**

Dear reviewers, we appreciate all your feedback and we will update the final version of the paper to accommodate your suggestions. We have addressed individual questions in replies to each reviewer, and we here provide a few general comments on our experimental setup and additional results in response to reviews. We seek to maintain a high level of transparency and very much welcome further discussion with each of the reviewers to address any remaining concerns that you may have.

**Experimental setup**

We would like to emphasize that, while we implement our method using DrQ and RAD as base algorithms, our method is agnostic to the choice of base algorithm and can trivially be extended to other deep Q-learning methods; both with and without learned policies, and for both continuous and discrete action spaces with minimal changes to the underlying algorithm. For fair comparison, we compare our method to DrQ with and without strong augmentation (as well as 4 other strong baselines) **with all methods using the same hyperparameters and network architecture**. See Section 8 in supplementary material for extensive details on our experimental setup, as well as pseudo-code for each variant of DrQ and SVEA. We have chosen to use DrQ with K=1, M=1 as the base algorithm for SVEA as higher values add a large computational overhead. Empirically, we observe approx. 6x wall-time for K=4, M=4 vs. K=1, M=1. We provide preliminary results for K=2,M=2 on the *walker, walk* task here*, and we will provide a more rigorous comparison with varying K,M for the DrQ baselines on the full set of tasks in the final version of the paper. Lastly, we would like to emphasize that our code will be made publicly available.

**Q-target variance**

We evaluate Q-target variance of SVEA and the DrQ + conv baseline in three tasks from DMControl and across 3 seeds, i.e. curve is mean variance and shaded area is +-1 std. deviation across seeds. Results are shown here**. Variance is continuously estimated from multiple forward passes with the target Q-function using the same observation. For DrQ + conv, this amounts to Q-value estimations on the same observations using different augmentations; for SVEA, no (strong) augmentation is applied to the target which dramatically reduces variance, but some variance remains due to action sampling from the policy. We observe a 640x reduction in variance in the *cartpole, swingup* task, and 18x and 31x reduction in the *walker, walk* and *walker, stand* tasks, respectively (measured at 300k steps). We will add a more rigorous evaluation of Q-target variance in the final version of the paper.

\* https://drive.google.com/file/d/1dJivNM-ce5Vn1QaLARYBqkIAtx6D9RAa/view?usp=sharing

\*\* https://drive.google.com/file/d/1_QCA7FxBDkemqKQVWi3e_wFmVBfox64m/view?usp=sharing

---

### Decision · Program_Chairs · 2021-09-27

**Decision:**

Accept (Poster)

**Comment:**

This paper investigates the problems related to the high variance in the targets that arise in TD-learning when the network is trained with data augmentation. Thee paper proposes a simple and effective way to address this problem by applying the data augmentation only on the online Q-network not on the target Qs and a modified objective function. The paper presents results on a range of tasks.

The paper is well-written and it is well-motivated. The authors investigate the proposed method on a range of interesting tasks and provide a very simple but seemingly effective solution.

The authors have provided an extensive rebuttal during the discussion period. However, some of the reviewers complaints are not completely addressed and I would appreciate if they can be addressed in the camera-ready version of this paper:

1) The theoretical justification provided in this paper is not rigorous enough. There is no guarantee that the proposed approach would work on any type of data augmentations. It would be nice if the authors are more careful about this and highlight that in the paper in order to avoid over-claiming.
2) The Reviewer MseA, xDk5 and UP5u asked several questions about the hyperparameters of the model. It would be nice if the authors can provide more analysis on the impact of the different hyperparameters used for different baselines. For example Reviewer UP5u pointed the different hyperparameters of DrQ and suggested more experiments to compare against.
3) More clear descriptions of the experimental setup along with a table of all hyperparameters used for all models (potentially in the appendix).
4) Adding error bars to the figures where applicable.

In addition to those points, I would recommend the authors to address all the other concerns by the reviewers as much as possible.